# Towards Lower Bounds on the Depth of ReLU Neural Networks

**Christoph Hertrich**
Technische Universität Berlin
Berlin, Germany
`christoph.hertrich@posteo.de`

**Amitabh Basu**
Johns Hopkins University
Baltimore, USA
`basu.amitabh@jhu.edu`

**Marco Di Summa**
Università degli Studi di Padova
Padua, Italy
`disumma@math.unipd.it`

**Martin Skutella**
Technische Universität Berlin
Berlin, Germany
`martin.skutella@tu-berlin.de`

## Abstract

We contribute to a better understanding of the class of functions that is represented by a neural network with ReLU activations and a given architecture. Using techniques from mixed-integer optimization, polyhedral theory, and tropical geometry, we provide a mathematical counterbalance to the universal approximation theorems which suggest that a single hidden layer is sufficient for learning tasks. In particular, we investigate whether the class of *exactly* representable functions *strictly* increases by adding more layers (with no restrictions on size). This problem has potential impact on algorithmic and statistical aspects because of the insight it provides into the class of functions represented by neural hypothesis classes. However, to the best of our knowledge, this question has not been investigated in the neural network literature. We also present upper bounds on the sizes of neural networks required to represent functions in these neural hypothesis classes.

## 1 Introduction

A core problem in machine learning or statistical pattern recognition is the estimation of an unknown data distribution with access to i.i.d. samples from the distribution. It is well-known that there is a tension between how much prior information one has about the data distribution and how many samples one needs to solve the problem with high confidence (or equivalently, how much variance one has in one's estimate). This is referred to as the *bias-variance* trade-off or the *bias-complexity* trade-off. Neural networks provide a way to turn this bias/complexity knob in a controlled manner that has been studied for decades going back to the idea of a *perceptron* by Rosenblatt [1958]. This is done by modifying the *architecture* of a neural network class of functions, which has two parameters: *depth* and *size*. As one increases these parameters, the class of functions becomes more expressive. In terms of the bias-variance trade-off, the "bias" decreases as the class of functions becomes more expressive, but the "variance" or "complexity" increases.

So-called *universal approximation theorems* [Cybenko, 1989, Hornik, 1991, Anthony and Bartlett, 1999] show that even with a single hidden layer, i.e., when the depth of the architecture is the smallest possible value, one can essentially reduce the "bias" as much as one desires, by increasing the size of the network or the number of neurons used in the neural network. Nevertheless, it can be advantageous both theoretically and empirically to increase the depth because a substantial reduction in the size can be achieved; Telgarsky [2016a], Eldan and Shamir [2016], Arora et al. [2018] is a small sample of recent work in this direction. To get a better quantitative handle on these trade-offs,

35th Conference on Neural Information Processing Systems (NeurIPS 2021).

it is important to understand what classes of functions are exactly representable by neural networks with a certain architecture. The precise mathematical statements of universal approximation theorems show that single layer networks can *approximate* arbitrarily well any continuous function (under some additional mild hypotheses). While this suggests that single layer networks are good enough from a learning perspective, from a mathematical perspective, one can ask the question if the class of functions represented by a single layer is a *strict* subset of the class of function represented by two or more hidden layers. On the question of size, one can ask for precise bounds on the size of the network of a given depth to represent a certain class of functions. We believe that a better understanding of the function classes exactly represented by different architectures will have implications not just for mathematical foundations, but also algorithmic and statistical learning aspects of neural networks. The task of searching for the "best" function in that class can only benefit from a better understanding of the nature of functions in that class. A motivating question behind the results in this paper is to understand the hierarchy of function classes exactly represented by neural networks of increasing depth.

We now introduce more precise notation and terminology to set the stage for our investigations.

**Notation.** We write $[n] := \{1, 2, \ldots, n\}$ for the set of natural numbers up to $n$ (without zero) and $[n]_0 := [n] \cup \{0\}$ for the same set including zero. For any $n \in \mathbb{N}$, let $\sigma \colon \mathbb{R}^n \to \mathbb{R}^n$ be the component-wise *rectifier* function

$$\sigma(x) = (\max\{0, x_1\}, \max\{0, x_2\}, \ldots, \max\{0, x_n\}).$$

For any *number of hidden layers* $k \in \mathbb{N}$, a $(k+1)$-*layer feedforward neural network with rectified linear units* (ReLU NN or simply NN) is given by $k$ affine transformations $T^{(\ell)} \colon \mathbb{R}^{n_{\ell-1}} \to \mathbb{R}^{n_\ell}$, $x \mapsto A^{(\ell)} x + b^{(\ell)}$, for $\ell \in [k]$, and a linear transformation $T^{(k+1)} \colon \mathbb{R}^{n_k} \to \mathbb{R}^{n_{k+1}}$, $x \mapsto A^{(k+1)} x$. It is said to *compute* or *represent* the function $f \colon \mathbb{R}^{n_0} \to \mathbb{R}^{n_{k+1}}$ given by

$$f = T^{(k+1)} \circ \sigma \circ T^{(k)} \circ \sigma \circ \cdots \circ T^{(2)} \circ \sigma \circ T^{(1)}.$$

The matrices $A^{(\ell)} \in \mathbb{R}^{n_\ell \times n_{\ell-1}}$ are called the *weights* and the vectors $b^{(\ell)} \in \mathbb{R}^{n_\ell}$ are the *biases* of the $\ell$-th layer. The number $n_\ell \in \mathbb{N}$ is called the *width* of the $\ell$-th layer. The maximum width of all hidden layers $\max_{\ell \in [k]} n_\ell$ is called the *width* of the NN. Further, we say that the NN has *depth* $k+1$ and *size* $\sum_{\ell=1}^{k} n_\ell$.

Often, NNs are represented as layered, directed, acyclic graphs where each dimension of each layer (including input layer $\ell = 0$ and output layer $\ell = k+1$) is one vertex, weights are arc labels, and biases are node labels. Then, the vertices are called *neurons*.

For a given input $x = x^{(0)} \in \mathbb{R}^{n_0}$, let $y^{(\ell)} := T^{(\ell)}(x^{(\ell-1)}) \in \mathbb{R}^{n_\ell}$ be the *activation vector* and $x^{(\ell)} := \sigma(y^\ell) \in \mathbb{R}^{n_\ell}$ the *output vector* of the $\ell$-th layer. Further, let $y := y^{(k+1)} = f(x)$ be the *output* of the NN. We also say that the $i$-th component of each of these vectors is the *activation* or the *output* of the $i$-th neuron in the $\ell$-th layer.

For $k \in \mathbb{N}$, we define

$$\text{ReLU}_n(k) := \{f \colon \mathbb{R}^n \to \mathbb{R} \mid f \text{ can be represented by a } (k+1)\text{-layer NN}\},$$
$$\text{CPWL}_n := \{f \colon \mathbb{R}^n \to \mathbb{R} \mid f \text{ is continuous and piecewise linear}\}.$$

By definition, a continuous function $f \colon \mathbb{R}^n \to \mathbb{R}$ is piecewise linear in case there is a finite set of polyhedra whose union is $\mathbb{R}^n$, and $f$ is affine linear over each such polyhedron.

In order to analyze $\text{ReLU}_n(k)$, we use another function class defined as follows. We call a function $g$ a $p$-*term max* function if it can be expressed as maximum of $p$ affine terms, that is, $g(x) = \max\{\ell_1(x), \ldots, \ell_p(x)\}$ where $\ell_i \colon \mathbb{R}^n \to \mathbb{R}$ is affinely linear for $i \in [p]$. Based on that, we define

$$\text{MAX}_n(p) := \{f \colon \mathbb{R}^n \to \mathbb{R} \mid f \text{ is a linear combination of } p\text{-term max functions}\}.$$

If the input dimension $n$ is not important for the context, we sometimes drop the index and use $\text{ReLU}(k) := \bigcup_{n \in \mathbb{N}} \text{ReLU}_n(k)$ and $\text{MAX}(p) := \bigcup_{n \in \mathbb{N}} \text{MAX}_n(p)$ instead.

Since we deal with polyhedra a lot in this paper, we will use the standard notations $\text{conv } A$ and $\text{cone } A$ for the convex and conic hulls of a set $A \subseteq \mathbb{R}^n$. For an in-depth treatment of polyhedra and (mixed-integer) optimization, we refer to Schrijver [1986].

## 1.1 Our Contribution

It is not hard to see that any function expressed by a ReLU network is a continuous and piecewise linear (CPWL) function, because one is composing continuous piecewise linear functions together. Based on a result by Wang and Sun [2005], Arora et al. [2018] show that every CPWL function defined on $\mathbb{R}^n$ can be represented by a ReLU neural network with $\lceil \log_2(n+1) \rceil$ hidden layers. We wish to understand whether one can do better. We believe it is not possible to do better and we pose the following conjecture to better understand the importance of depth in neural networks.

**Conjecture 1.1.** *For any $n \in \mathbb{N}$, let $k^* := \lceil \log_2(n+1) \rceil$. Then it holds that*

$$\mathrm{ReLU}_n(0) \subsetneq \mathrm{ReLU}_n(1) \subsetneq \cdots \subsetneq \mathrm{ReLU}_n(k^*-1) \subsetneq \mathrm{ReLU}_n(k^*) = \mathrm{CPWL}_n. \qquad (1)$$

Conjecture 1.1 claims that any additional layer up to $k^*$ hidden layers strictly increases the set of representable functions. This would imply that the construction by Arora et al. [2018, Theorem 2.1] is actually depth-minimal.

Observe that in order to prove Conjecture 1.1, it suffices to find a single function $f \in \mathrm{ReLU}_n(k^*) \setminus \mathrm{ReLU}_n(k^*-1)$ with $n = 2^{k^*-1}$ for all $k^* \in \mathbb{N}$. This also implies all remaining strict inclusions $\mathrm{ReLU}_n(i-1) \subsetneq \mathrm{ReLU}_n(i)$ for $i < k^*$ since $\mathrm{ReLU}_n(i-1) = \mathrm{ReLU}_n(i)$ directly implies that $\mathrm{ReLU}_n(i-1) = \mathrm{ReLU}_n(i')$ for all $i' \geq i - 1$.

In fact, there is a canonical candidate for such a function, allowing us to reformulate the conjecture as follows.

**Conjecture 1.2.** *For any $k \in \mathbb{N}$, $n = 2^k$, the function $f_n(x) = \max\{0, x_1, \ldots, x_n\}$ cannot be represented with $k$ hidden layers.*

**Proposition 1.3.** *Conjecture 1.1 and Conjecture 1.2 are equivalent.*

*Proof (Sketch).* We argued above that Conjecture 1.2 implies Conjecture 1.1. For the other direction, one can argue that, if the specific $(n+1)$-term max function $f_n$ can be represented by $k$ hidden layers, then every other $(n+1)$-term max function as well. The claim then follows via a result by [Wang and Sun, 2005] stating that any $f \in \mathrm{CPWL}_n$ can be written as linear combination of $(n+1)$-term max functions. We provide a detailed argument in Appendix A. $\qquad \square$

It is known that Conjecture 1.2 holds for $k = 1$ [Mukherjee and Basu, 2017]. However, the conjecture remains open for $k \geq 2$. In this paper, we present the following results as partial progress towards resolving this conjecture.

In Section 2, we resolve Conjecture 1.2 for $k = 2$, under a natural assumption on the breakpoints of the function represented by any intermediate neuron. We achieve this result by leveraging techniques from mixed-integer programming to analyze the set of functions computable by certain NNs.

It is not hard to see that $\mathrm{MAX}(2^k) \subseteq \mathrm{ReLU}(k)$ for all $k \in \mathbb{N}$ [Arora et al., 2018], that is, any $2^k$-term max function (and linear combinations thereof) can be expressed with $k$ hidden layers. One might ask whether the converse is true as well, that is, whether the classes $\mathrm{MAX}(2^k)$ and $\mathrm{ReLU}(k)$ are actually equal. This would not only provide a neat characterization of $\mathrm{ReLU}(k)$, but also prove Conjecture 1.2 without any additional assumption since one can show that $\max\{0, x_1, \ldots, x_{2^k}\}$ is not contained in $\mathrm{MAX}(2^k)$.

In fact, this is true for $k = 1$, that is, a function is computable with one hidden layer if and only if it is a linear combination of 2-term max functions. However, in Section 3, we show that for $k \geq 2$, the class $\mathrm{ReLU}(k)$ is a strict superset of $\mathrm{MAX}(2^k)$. In this section, the key technical ingredient is the theory of polyhedral complexes associated with CPWL functions. This way, we provide important insights concerning the richness of the class $\mathrm{ReLU}(k)$.

So far, we have focused on understanding the smallest depth needed to express CPWL functions using neural networks with ReLU activations. In Section 4, we complement these results by upper bounds on the sizes of the networks needed for expressing arbitrary CPWL functions. In particular, Theorem 4.4 shows that any continuous piecewise linear function with $p$ linear/affine pieces on $\mathbb{R}^n$ can be expressed by a network with depth at most $\mathcal{O}(\log n)$ and width at most $p^{\mathcal{O}(n^2)}$. We arrive at this result by introducing a novel application of recently established interactions between neural networks and tropical geometry.

## 1.2 Related Work

**Depth versus size.**   Soon after the original universal approximation theorems [Cybenko, 1989, Hornik, 1991], concrete bounds were obtained on the number of neurons needed in the hidden layer to achieve a certain level of accuracy. The literature on this is vast and we refer to a small representative sample here [Barron, 1993, 1994, Mhaskar, 1993, Pinkus, 1999, Mhaskar, 1996, Mhaskar and Micchelli, 1995]. More recently, work has focused on how deeper networks can have exponentially or super exponentially smaller size compared to shallower networks [Telgarsky, 2016a, Eldan and Shamir, 2016, Arora et al., 2018, Vardi et al., 2021]. See also Gribonval et al. [2021] for another perspective on the relationship between expressivity and architecture, and the references therein. We reiterate that the list of references above is far from complete.

**Mixed-integer optimization and machine learning.**   Over the past decade, a growing body of work has emerged that explores the interplay between mixed-integer optimization and machine learning. On the one hand, researchers have attempted to improve mixed-integer optimization algorithms by exploiting novel techniques from machine learning [Bonami et al., 2018, Gasse et al., 2019, He et al., 2014, Khalil et al., 2016, 2017, Kruber et al., 2017, Lodi and Zarpellon, 2017, Alvarez et al., 2017]; see also Bengio et al. [2020] for a recent survey. On the flip side, mixed-integer optimization techniques have been used to analyze function classes represented by neural networks [Serra et al., 2018, Anderson et al., 2020, Fischetti and Jo, 2017, Serra and Ramalingam, 2020, Serra et al., 2020]. In Section 2 below, we show another new use of mixed-integer optimization tools for understanding function classes represented by neural networks.

**Design of training algorithms.**   We believe that a better understanding of the function classes represented exactly by a neural architecture also has benefits in terms of understanding the complexity of the training problem. For instance, in a paper by Arora et al. [2018], an understanding of single layer ReLU networks enables the design of a globally optimal algorithm for solving the empirical risk minimization (ERM) problem, that runs in polynomial time in the number of data points in fixed dimension. See also Goel et al. [2017, 2018], Goel and Klivans [2019], Dey et al. [2020], Boob et al. [2020], Goel et al. [2021], Froese et al. [2021] for a similar lines of work.

**Neural Networks and Tropical Geometry.**   A recent stream of research involves the interplay between neural networks and tropical geometry. The piecewise linear functions computed by neural networks can be seen as (tropical quotients of) tropical polynomials. Linear regions of these functions correspond to vertices of so-called *Newton polytopes* associated with these tropical polynomials. Applications of this correspondence include bounding the number of linear regions of a neural network [Zhang et al., 2018, Charisopoulos and Maragos, 2018, Montúfar et al., 2021] and understanding decision boundaries [Alfarra et al., 2020]. In Section 4 we present a novel application of tropical concepts to understand neural networks. We refer to Maragos et al. [2021] for a recent survey of connections between machine learning and tropical geometry, as well as to the textbooks by Maclagan and Sturmfels [2015] and Joswig [2022] for in-depth introductions to tropical geometry and tropical combinatorics.

## 2  Conditional Lower Bounds on Depth via Mixed-Integer Programming

In this section, we provide a computer-aided proof that, under a natural, yet unproven assumption, the function $f(x) \coloneqq \max\{0, x_1, x_2, x_3, x_4\}$ cannot be represented by a 3-layer NN. It is worth to note that, to the best of our knowledge, no CPWL function is known for which the non-existence of a 3-layer NN can be proven without additional assumption. For easier notation, we write $x_0 \coloneqq 0$.

We first prove that we may restrict ourselves to NNs without biases. This holds true independent of our assumption, which we introduce afterwards.

**Definition 2.1.** A function $g \colon \mathbb{R}^n \to \mathbb{R}^m$ is called *positively homogeneous* if $g(\lambda x) = \lambda g(x)$ for all $\lambda \geq 0$.

**Definition 2.2.** For an NN given by affine transformations $T^{(\ell)}(x) = A^{(\ell)}x + b^{(\ell)}$, we define the corresponding *homogenized NN* to be the NN given by $\tilde{T}^{(\ell)}(x) = A^{(\ell)}x$ with all biases set to zero.

**Proposition 2.3.** *If an NN computes a positively homogeneous function, then the corresponding homogenized NN computes the same function.*

*Proof.* Let $g \colon \mathbb{R}^{n_0} \to \mathbb{R}^{n_{k+1}}$ be the function computed by the original NN and $\tilde{g}$ the one computed by the homogenized NN. Further, for any $0 \leq \ell \leq k$, let $g^{(\ell)} = T^{(\ell+1)} \circ \sigma \circ T^{(\ell)} \circ \cdots \circ T^{(2)} \circ \sigma \circ T^{(1)}$ be the function computed by the sub-NN consisting of the first $(\ell + 1)$-layers and let $\tilde{g}^{(\ell)}$ be the function computed by the corresponding homogenized sub-NN. We first show by induction on $\ell$ that the norm of $\|g^{(\ell)}(x) - \tilde{g}^{(\ell)}(x)\|$ is bounded by a global constant that only depends on the parameters of the NN but not on $x$.

For $\ell = 0$, we obviously have $\|g^{(0)}(x) - \tilde{g}^{(0)}(x)\| = \|b^{(1)}\| =: C_0$, settling the induction base. For the induction step, let $\ell \geq 1$ and assume that $\|g^{(\ell-1)}(x) - \tilde{g}^{(\ell-1)}(x)\| \leq C_{\ell-1}$, where $C_{\ell-1}$ only depends on the parameters of the NN. Since component-wise ReLU activation has Lipschitz constant 1, this implies $\|(\sigma \circ g^{(\ell-1)})(x) - (\sigma \circ \tilde{g}^{(\ell-1)})(x)\| \leq C_{\ell-1}$. Using any matrix norm that is compatible with the Euclidean vector norm, we obtain:

$$
\begin{aligned}
\|g^{(\ell)}(x) - \tilde{g}^{(\ell)}(x)\| &= \|b^{(\ell+1)} + A^{(\ell+1)}((\sigma \circ g^{(\ell-1)})(x) - (\sigma \circ \tilde{g}^{(\ell-1)})(x))\| \\
&\leq \|b^{(\ell+1)}\| + \|A^{(\ell+1)}\| \cdot C_{\ell-1} =: C_\ell
\end{aligned}
$$

Since the right-hand side only depends on NN parameters, the induction is completed.

Finally, we show that $g = \tilde{g}$. For the sake of contradiction, suppose that there is an $x \in \mathbb{R}^{n_0}$ with $\|g(x) - \tilde{g}(x)\| = \delta > 0$. Let $x' := \frac{C_k+1}{\delta} x$; then, by positive homogeneity, it follows that $\|g(x') - \tilde{g}(x')\| = C_k + 1 > C_k$, contradicting the property shown above. Thus, we have $g = \tilde{g}$. $\square$

Since $f = \max\{0, x_1, x_2, x_3, x_4\}$ is positively homogeneous, Proposition 2.3 implies that, if there is a 3-layer NN computing $f$, then there also is one that has no biases. Therefore, in the remainder of this section, we only consider NNs without biases and assume implicitly that all considered CPWL functions are positively homogeneous. In particular, any piece of such a CPWL function is linear and not only affine linear.

Observe that, for function $f$, the only points of non-differentiability (a.k.a. *breakpoints*) are at places where at least two of the five numbers $x_0 = 0$, $x_1$, $x_2$, $x_3$, and $x_4$ are equal. Hence, if some neuron of an NN computing $f$ introduces breakpoints at other places, these breakpoints must be canceled out by other neurons. Therefore, it is a natural assumption that such breakpoints are not introduced at all in the first place.

To make this assumption formal, let $H_{ij} = \{x \in \mathbb{R}^4 \mid x_i = x_j\}$, for $0 \leq i < j \leq 4$, be ten hyperplanes in $\mathbb{R}^4$ and $H = \bigcup_{0 \leq i < j \leq 4} H_{ij}$ be the corresponding hyperplane arrangement. The *regions* or *cells* of $H$ are defined to be the closures of the connected components of $\mathbb{R}^4 \setminus H$. It is easy to see that these regions are in one-to-one correspondence to the $5! = 120$ possible orderings of the five numbers $x_0 = 0$, $x_1$, $x_2$, $x_3$, and $x_4$. More precisely, for a permutation $\pi$ of the five indices $[4]_0 = \{0, 1, 2, 3, 4\}$, the corresponding region is the polyhedron

$$
C_\pi := \{x \in \mathbb{R}^4 \mid x_{\pi(0)} \leq x_{\pi(1)} \leq x_{\pi(2)} \leq x_{\pi(3)} \leq x_{\pi(4)}\}.
$$

We say that a (positively homogeneous) CPWL function $g$ is $H$-*conforming*, if it is linear within any of these regions of $H$, that is, if it only has breakpoints where the relative ordering of the five values $x_0 = 0$, $x_1$, $x_2$, $x_3$, $x_4$ changes. Moreover, an NN is said to be $H$-conforming if the output of each neuron contained in the NN is $H$-conforming. Equivalently, this is the case if and only if all intermediate functions $\sigma \circ T^{(\ell)} \circ \sigma \circ T^{(\ell-1)} \circ \cdots \circ \sigma \circ T^{(1)}$, $\ell \in [k]$, are $H$-conforming. Now our assumption can be formally phrased as follows.

**Assumption 2.4.** *If there exists a 3-layer NN computing $f(x) = \max\{0, x_1, x_2, x_3, x_4\}$, then there also exists one that is $H$-conforming.*

We use mixed-integer programming to prove the following theorem.

**Theorem 2.5.** *Under Assumption 2.4, there does not exist a 3-layer NN that computes the function $f(x) = \max\{0, x_1, x_2, x_3, x_4\}$.*

*Proof (Outline).* We first study some geometric properties of the hyperplane arrangement $H$. This will show that each of the 120 cells of $H$ is a simplicial polyhedral cone spanned by 4 extreme rays. In total, there are 30 such rays (because rays are used multiple times to span different cones). This implies that the set of $H$-conforming functions of type $\mathbb{R}^4 \to \mathbb{R}$ is a 30-dimensional vector space and

each function is uniquely determined by its values on the 30 rays. We then use linear algebra to show that the space of functions generated by $H$-conforming two-layer NNs is a 14-dimensional subspace. Moreover, with two hidden layers, at least 29 of the 30 dimensions can be generated and $f$ is not contained in this 29-dimensional subspace. So, it remains the question whether the 14 dimensions producible with the first hidden layer can be combined in such a way that after applying a ReLU activation in the second hidden layer, we do not end up within the 29-dimensional subspace. We model this question as a mixed-interger program (MIP). Solving the MIP yields that we always end up within the 29-dimensional subspace, implying that $f$ cannot be represented by a 3-layer NN. This provides a computational proof of Theorem 2.5. All details can be found in Appendix B. ☐

## 3  Going Beyond $2^k$-Term Max Functions with $k$ Layers

In this section we prove the following result:

**Theorem 3.1.** *For any $k \geq 2$, the class* $\mathrm{ReLU}(k)$ *is a strict superset of* $\mathrm{MAX}(2^k)$.

In order to prove this theorem, we provide a specific function that is in $\mathrm{ReLU}(k) \setminus \mathrm{MAX}(2^k)$ for any number of hidden layers $k \geq 2$. The challenging part is to show that the function is in fact not contained in $\mathrm{MAX}(2^k)$.

**Proposition 3.2.** *For any $n \geq 3$, the function*

$$f \colon \mathbb{R}^n \to \mathbb{R}, \quad f(x) = \max\{0, x_1, x_2, \ldots, x_{n-3}, \max\{x_{n-2}, x_{n-1}\} + \max\{0, x_n\}\} \quad (2)$$

*cannot be written as a linear combination of $n$-term max functions.*

The above proposition means that it is not possible to write $f(x)$ in the form

$$f(x) = \sum_{i=1}^{p} \lambda_i \max\{\ell_{i1}(x), \ldots, \ell_{in}(x)\}$$

where $p \in \mathbb{N}$, $\lambda_1, \ldots, \lambda_p \in \mathbb{R}$, and $\ell_{ij} : \mathbb{R}^n \to \mathbb{R}$ is an affine linear function for every $i \in [p]$ and $j \in [n]$. (Note that max functions with less than $n$ terms are also allowed, as some functions $\ell_{ij}$ may coincide.)

Before we prove Proposition 3.2, we show that it implies Theorem 3.1.

*Proof of Theorem 3.1.* For $k \geq 2$, let $n := 2^k$. By Proposition 3.2, function $f$ defined in (2) is not contained in $\mathrm{MAX}(2^k)$. It remains to show that it can be represented using a ReLU NN with $k$ hidden layers. To see this, first observe that any of the $n/2 = 2^{k-1}$ terms $\max\{0, x_1\}$, $\max\{x_{2i}, x_{2i+1}\}$ for $i \in [n/2 - 2]$, and $\max\{x_{n-2}, x_{n-1}\} + \max\{0, x_n\}$ can be expressed by a one-hidden-layer NN since all these are (linear combinations of) 2-term max functions. Since $f$ is the maximum of these $2^{k-1}$ terms, and since the maximum of $2^{k-1}$ numbers can be computed with $k - 1$ hidden layers [Arora et al., 2018], this implies that $f$ is in $\mathrm{ReLU}(k)$. ☐

In order to prove Proposition 3.2, we need the concept of polyhedral complexes. A *polyhedral complex* $\mathcal{P}$ is a finite set of polyhedra such that each face of a polyhedron in $\mathcal{P}$ is also in $\mathcal{P}$, and for two polyhedra $P, Q \in \mathcal{P}$, their intersection $P \cap Q$ is a common face of $P$ and $Q$ (possibly the empty face). Given a polyhedral complex $\mathcal{P}$ in $\mathbb{R}^n$ and an integer $m \in [n]$, we let $\mathcal{P}^m$ denote the collection of all $m$-dimensional polyhedra in $\mathcal{P}$.

For a convex CPWL function $f$, we define its *underlying polyhedral complex* as follows: it is the unique polyhedral complex covering $\mathbb{R}^n$ (i.e., each point in $\mathbb{R}^n$ belongs to some polyhedron in $\mathcal{P}$) whose $n$-dimensional polyhedra coincide with the domains of the (maximal) affine pieces of $f$. In particular, $f$ is affinely linear within each $P \in \mathcal{P}$, but not within any strict superset of a polyhedron in $\mathcal{P}^n$.

Exploiting properties of polyhedral complexes associated with CPWL functions, we prove the following proposition in Appendix C.

**Proposition 3.3.** *Let $f_0 \colon \mathbb{R}^n \to \mathbb{R}$ be a convex CPWL function and let $\mathcal{P}_0$ be the underlying polyhedral complex. If there exists a hyperplane $H \subseteq \mathbb{R}^n$ such that the set*

$$T := \bigcup \{F \in \mathcal{P}_0^{n-1} \mid F \subseteq H\}$$

*is nonempty and contains no line, then $f_0$ cannot be expressed as a linear combination of $n$-term maxima of affine linear functions.*

This allows us to prove Proposition 3.2.

*Proof of Proposition 3.2.* Observe that $f$ (defined in (2)) has the alternate representation

$$f(x) = \max\{0, \, x_1, \, x_2, \, \ldots, \, x_{n-3}, \, x_{n-2}, \, x_{n-1}, \, x_{n-2} + x_n, \, x_{n-1} + x_n\}$$

as a maximum of $n + 2$ terms. Let $\mathcal{P}$ be its underlying polyhedral complex. Let the hyperplane $H$ be defined by $x_1 = 0$.

Observe that any facet in $\mathcal{P}^{n-1}$ is a polyhedron defined by two of the $n + 2$ terms that are equal and at least as large as each of the remaining $n$ terms. Hence, the only facet that could possibly be contained in $H$ is

$$F := \{x \in \mathbb{R}^n \mid x_1 = 0 \geq x_2, \, \ldots, \, x_{n-3}, \, x_{n-2}, \, x_{n-1}, \, x_{n-2} + x_n, \, x_{n-1} + x_n\}.$$

Note that $F$ is indeed an $(n-1)$-dimensional facet in $\mathcal{P}^{n-1}$, because, for example, the full neighborhood of $(0, -1, \ldots, -1) \in \mathbb{R}^n$ intersected with $H$ is contained in $F$.

Finally, we need to show that $F$ is pointed, that is, it contains no line. A well-known fact from polyhedral theory says if there is any line in $F$ with direction $d \in \mathbb{R}^n \setminus \{0\}$, then $d$ must satisfy the defining inequalities with equality. However, only the zero vector does this. Hence, $F$ cannot contain a line.

Therefore, when applying Proposition 3.3 to $f$ with underlying polyhedral complex $\mathcal{P}$ and hyperplane $H$, we have $T = F$, which is nonempty and contains no line. Hence, $f$ cannot be written as linear combination of $n$-term maxima. $\square$

## 4   A Width Bound for NNs with Small Depth

While the arguments in Arora et al. [2018] show that $\mathrm{CPWL}_n = \mathrm{ReLU}_n(\lceil \log_2(n + 1) \rceil)$, they do not provide any bound on the width of the NN required to represent any particular continuous piecewise linear function. The purpose of this section is to prove that for fixed dimension $n$, the required width for exact, depth-minimal representation of a CPWL function can be polynomially bounded in the number $p$ of affine pieces; in particular $p^{\mathcal{O}(n^2)}$. This is closely related to works that bound the number of linear pieces of an NN as a function of the size [Montúfar et al., 2014, Pascanu et al., 2014, Raghu et al., 2017, Montúfar et al., 2021]. It can also be seen as a counterpart, in the context of exact representations, to quantitative universal approximation theorems that bound the number of neurons required to achieve a certain approximation guarantee; see, e.g., Barron [1993, 1994], Mhaskar [1993], Pinkus [1999], Mhaskar [1996], Mhaskar and Micchelli [1995].

### 4.1   The Convex Case

We first derive our result for the case of convex CPWL functions and then use this to also prove the general nonconvex case. Our width bound is a consequence of the following theorem about convex CPWL functions, for which we are going to provide a geometric proof later.

**Theorem 4.1.** *Let $f(x) = \max\{a_i^T x + b_i \mid i \in [p]\}$ be a convex CPWL function defined on $\mathbb{R}^n$. Then $f$ can be written as*

$$f(x) = \sum_{\substack{S \subseteq [p], \\ |S| \leq n+1}} c_S \max\{a_i^T x + b_i \mid i \in S\}$$

*with coefficients $c_S \in \mathbb{Z}$, for $S \subseteq [p]$, $|S| \leq n + 1$.*

For the convex case, this yields a stronger version of the theorem by Wang and Sun [2005] stating that any (not necessarily convex) CPWL function can be written as a linear combination of $(n + 1)$-term maxima. Theorem 4.1 is stronger in the sense that it guarantees that all pieces of the $(n + 1)$-term maxima must be pieces of the original function, making it possible to bound the total number of these $(n + 1)$-term maxima and, therefore, the size of an NN representing $f$.

**Theorem 4.2.** *Let $f \colon \mathbb{R}^n \to \mathbb{R}$ be a convex CPWL function with $p$ affine pieces. Then $f$ can be represented by a ReLU NN with depth $\lceil \log_2(n+1) \rceil + 1$ and width $\mathcal{O}(p^{n+1})$.*

*Proof.* Since the number of possible subsets $S \subseteq [p]$ with $|S| \leq n+1$ is bounded by $p^{n+1}$, the theorem follows by Theorem 4.1 and the construction from Arora et al. [2018, Theorem 2.1]. $\qquad\square$

Before we present the proof of Theorem 4.1, we show how we can generalize its consequences to the nonconvex case.

## 4.2 The General (Nonconvex) Case

It is a well-known fact that every CPWL function can be expressed as a difference of two convex CPWL functions, see, e.g., Wang [2004, Theorem 1]. This allows us to derive the general case from the convex case. What we need, however, is to bound the number of affine pieces of the two convex CPWL functions in terms of the number of pieces of the original function. Therefore, we consider a specific decomposition for which such bounds can easily be achieved.

**Proposition 4.3.** *Let $f \colon \mathbb{R}^n \to \mathbb{R}$ be a CPWL function with $p$ affine pieces. Then, $f$ can be written as $f = g - h$ where both $g$ and $h$ are convex CPWL functions with at most $p^{2n+1}$ pieces.*

*Proof.* Suppose the $p$ affine pieces of $f$ are given by $x \mapsto a_i^T x + b_i$, $i \in [p]$. Define the function $h(x) := \sum_{1 \leq i < j \leq p} \max\{a_i^T x + b_i, a_j^T x + b_j\}$ and let $g := f + h$. Then, obviously, $f = g - h$. It remains to show that both $g$ and $h$ are convex CPWL functions with at most $p^{2n+1}$ pieces.

The convexity of $h$ is clear by definition. Consider the $\binom{p}{2} = \frac{p(p-1)}{2} < p^2$ hyperplanes given by $a_i^T x + b_i = a_j^T x + b_j$, $1 \leq i < j \leq p$. They divide $\mathbb{R}^n$ into at most $\binom{p^2}{n} + \binom{p^2}{n-1} + \cdots + \binom{p^2}{0} \leq p^{2n}$ regions (compare Edelsbrunner [1987, Theorem 1.3]) in each of which $h$ is affine. In particular, $h$ has at most $p^{2n} \leq p^{2n+1}$ pieces.

Next, we show that $g = f + h$ is convex. Intuitively, this holds because each possible breaking hyperplane of $f$ is made convex by adding $h$. To make this formal, note that by the definition of convexity, it suffices to show that $g$ is convex along each affine line. For this purpose, consider an arbitrary line $x(t) = ta + b$, $t \in \mathbb{R}$, given by $a \in \mathbb{R}^n$ and $b \in \mathbb{R}$. Let $\tilde{f}(t) := f(x(t))$, $\tilde{g}(t) := g(x(t))$, and $\tilde{h}(t) := h(x(t))$. We need to show that $\tilde{g} \colon \mathbb{R} \to \mathbb{R}$ is a convex function. Observe that $\tilde{f}, \tilde{g}$, and $\tilde{h}$ are clearly one-dimensional CPWL functions with the property $\tilde{g} = \tilde{f} + \tilde{h}$. Hence, it suffices to show that $\tilde{g}$ is convex locally around each of its breakpoints. Let $t \in \mathbb{R}$ be an arbitrary breakpoint of $\tilde{g}$. If $\tilde{f}$ is already convex locally around $t$, then the same holds for $\tilde{g}$ as well since $\tilde{h}$ inherits convexity from $h$. Now suppose that $t$ is a nonconvex breakpoint of $\tilde{f}$. Then there exist two distinct pieces of $f$, indexed by $i, j \in [p]$ with $i \neq j$, such that $\tilde{f}(t') = \min\{a_i^T x(t') + b_i, a_j^T x(t') + b_j\}$ for all $t'$ sufficiently close to $t$. By construction, $\tilde{h}(t')$ contains the summand $\max\{a_i^T x(t') + b_i, a_j^T x(t') + b_j\}$. Thus, adding this summand to $\tilde{f}$ linearizes the nonconvex breakpoint of $\tilde{f}$, while adding all the other summands preserves convexity. In total, $\tilde{g}$ is convex locally around $t$, which finishes the proof that $g$ is a convex function.

Finally, observe that pieces of $g = f + h$ are always intersections of pieces of $f$ and $h$, for which we have only $p \cdot p^{2n} = p^{2n+1}$ possibilities. $\qquad\square$

Having this, we may conclude the following.

**Theorem 4.4.** *Let $f \colon \mathbb{R}^n \to \mathbb{R}$ be a CPWL function with $p$ affine pieces. Then $f$ can be represented by a ReLU NN with depth $\lceil \log_2(n+1) \rceil + 1$ and width $\mathcal{O}(p^{2n^2+3n+1})$.*

*Proof.* Consider the decomposition $f = g - h$ from Proposition 4.3. Using Theorem 4.2, we obtain that both $g$ and $h$ can be represented with the required depth $\lceil \log_2(n+1) \rceil + 1$ and with width $\mathcal{O}((p^{2n+1})^{n+1}) = \mathcal{O}(p^{2n^2+3n+1})$. Thus, the same holds for $f$. $\qquad\square$

## 4.3 Extended Newton Polyhedra of Convex CPWL Functions

For our proof of Theorem 4.1, we use a correspondence of convex CPWL functions with certain polyhedra, which are known as (extended) Newton polyhedra in tropical geometry [Maclagan and Sturmfels, 2015]. These relations between tropical geometry and neural networks have previously been applied to investigate expressivity of NNs; compare our references in the introduction.

In order to formalize this correspondence, let $\mathrm{CCPWL}_n \subseteq \mathrm{CPWL}_n$ be the set of convex CPWL functions of type $\mathbb{R}^n \to \mathbb{R}$. For $f(x) = \max\{a_i^T x + b_i \mid i \in [p]\}$ in $\mathrm{CCPWL}_n$, we define its so-called *extended Newton polyhedron* to be

$$\mathcal{N}(f) := \mathrm{conv}(\{(a_i, b_i) \in \mathbb{R}^n \times \mathbb{R} \mid i \in [p]\}) + \mathrm{cone}(\{-e_{n+1}\}) \subseteq \mathbb{R}^{n+1}.$$

We denote the set of all possible extended Newton polyhedra in $\mathbb{R}^{n+1}$ as $\mathrm{Newt}_n$. That is, $\mathrm{Newt}_n$ is the set of (unbounded) polyhedra in $\mathbb{R}^{n+1}$ that emerge from a polytope by adding the negative of the $(n + 1)$-st unit vector $-e_{n+1}$ as an extreme ray. Hence, a set $P \subseteq \mathbb{R}^{n+1}$ is an element of $\mathrm{Newt}_n$ if and only if $P$ can be written as $P = \mathrm{conv}(\{(a_i, b_i) \in \mathbb{R}^n \times \mathbb{R} \mid i \in [p]\}) + \mathrm{cone}(\{-e_{n+1}\})$. Conversely, for a polyhedron $P \in \mathrm{Newt}_n$ of this form, let $\mathcal{F}(P) \in \mathrm{CCPWL}_n$ be the function defined by $\mathcal{F}(P)(x) = \max\{a_i^T x + b_i \mid i \in [p]\}$.

There is an intuitive way of thinking about the extended Newton polyhedron $P$ of a convex CPWL function $f$: it consists of all hyperplane coefficients $(a, b) \in \mathbb{R}^n \times \mathbb{R}$ such that $a^T x + b \leq f(x)$ for all $x \in \mathbb{R}^n$. This also explains why we add the extreme ray $-e_{n+1}$: decreasing $b$ obviously maintains the property of $a^T x + b$ being a lower bound on the function $f$.

We need the notion of the *Minkowski sum* of two polyhedra $P$ and $Q$: it is given as the set $P + Q = \{p + q \mid p \in P, q \in Q\}$.

In fact, there is a one-to-one correspondence between elements of $\mathrm{CCPWL}_n$ and $\mathrm{Newt}_n$, which is nicely compatible with some (functional and polyhedral) operations. This correspondence has been studied before in tropical geometry [Maclagan and Sturmfels, 2015, Joswig, 2022], convex geometry[1] [Hiriart-Urruty and Lemaréchal, 1993], as well as neural network literature [Zhang et al., 2018, Charisopoulos and Maragos, 2018, Alfarra et al., 2020, Montúfar et al., 2021]. We summarize the key findings of this correspondence relevant to our work in the following proposition:

**Proposition 4.5.** *Let $n \in \mathbb{N}$ and $f_1, f_2 \in \mathrm{CCPWL}_n$. Then it holds that*

*(i) the functions $\mathcal{N} \colon \mathrm{CCPWL}_n \to \mathrm{Newt}_n$ and $\mathcal{F} \colon \mathrm{Newt}_n \to \mathrm{CCPWL}_n$ are well-defined, that is, their output is independent from the representation of the input by pieces or vertices, respectively,*

*(ii) $\mathcal{N}$ and $\mathcal{F}$ are bijections and inverse to each other,*

*(iii) $\mathcal{N}(\max\{f_1, f_2\}) = \mathrm{conv}(\mathcal{N}(f_1), \mathcal{N}(f_2)) := \mathrm{conv}(\mathcal{N}(f_1) \cup \mathcal{N}(f_2))$,*

*(iv) $\mathcal{N}(f_1 + f_2) = \mathcal{N}(f_1) + \mathcal{N}(f_2)$, where the $+$ on the right-hand side is Minkowski addition.*

An algebraic way of phrasing this proposition is as follows: $\mathcal{N}$ and $\mathcal{F}$ are isomorphisms between the semirings $(\mathrm{CCPWL}_n, \max, +)$ and $(\mathrm{Newt}_n, \mathrm{conv}, +)$.

## 4.4 Outline of the Proof of Theorem 4.1

We prove Theorem 4.1 in full detail in Appendix D. The rough idea is as follows.

Suppose we have a $p$-term max function $f$ with $p \geq n + 2$. By Proposition 4.5, $f$ corresponds to a polyhedron $P \in \mathrm{Newt}_n$ with at least $n + 2$ vertices. Applying a classical result from discrete geometry known as *Radon's theorem* allows us to carefully decompose $P$ into a "signed"[2] Minkowski sum of polyhedra in $\mathrm{Newt}_n$ whose vertices are subsets of at most $p - 1$ out of the $p$ vertices of $P$. Translating this back into the world of CPWL functions by Proposition 4.5 yields that $f$ can be written as linear combination of $p'$-term maxima with $p' < p$, where each of them involves a subset of

---

[1]$\mathcal{N}(f)$ is the negative of the epigraph of the convex conjugate of $f$.

[2]Some polyhedra may occur with "negative" coefficents in that sum, meaning that they are actually added to $P$ instead of the other polyhedra. The corresponding CPWL functions will then have negative coefficients in the linear combination representing $f$.

the $p$ affine terms of $f$. We can then obtain Theorem 4.1 by iterating until every occurring maximum expression involves at most $n + 1$ terms.

### 4.5 Potential Approaches to Show Lower Bounds on the Width

In light of the upper width bounds shown in this section, a natural question to ask is whether also meaningful lower bounds can be achieved. This would mean constructing a family of CPWL functions with $p$ pieces defined on $\mathbb{R}^n$ (with different values of $p$ and $n$), for which we can prove that a large width is required to represent these functions with NNs of depth $\lceil \log_2(n + 1) \rceil + 1$.

A trivial and not very satisfying answer follows, e.g., from Raghu et al. [2017] or Serra et al. [2018]: for fixed input dimension $n$, they show that a function computed by an NN with $k$ hidden layers and width $w$ has at most $\mathcal{O}(w^{kn})$ pieces. For our setting, this means that an NN with logarithmic depth needs a width of at least $\mathcal{O}(p^{1/(n \log n)})$ to represent a function with $p$ pieces. This is, of course, very far away from our upper bounds.

Similar upper bounds on the number of pieces have been proven by many other authors and are often used to show depth-width tradeoffs [Montúfar et al., 2014, 2021, Pascanu et al., 2014, Telgarsky, 2016b, Arora et al., 2018]. However, there is a good reason why all these results only give rise to very trivial lower bounds for our setting: the focus is always on functions with very many pieces, which then, consequently, need many neurons to be represented (with small depth). However, since the lower bounds we strive for depend on the number of pieces, we would need to construct a family of functions with comparably few pieces that still need very many neurons to be represented. In general, it seems to be a tough task to argue why such functions should exist.

A different approach could leverage methods from complexity theory, in particular from circuit complexity. Neural networks are basically arithmetic circuits with very special operations allowed. In fact, they can be seen as a tropical variant of arithmetic circuits. Showing circuit lower bounds is a notoriously difficult task in complexity theory, but maybe some conditional result (based on common conjectures similar to $P \neq NP$) could be established.

We think that the question whether our bounds are tight, or whether at least some non-trivial lower bounds on the width for NNs with logarithmic depth can be shown, is an exciting question for further research.

## 5 Discussion of Future Research Directions

The most obvious and, at the same time, most exciting open research question is to prove or disprove Conjecture 1.1, or equivalently Conjecture 1.2. The first step could be to prove Assumption 2.4. The assumption is intuitive because every breakpoint introduced at other places needs to be canceled out later. Therefore, it is natural to assume that these breakpoints do not have to be introduced in the first place. However, this intuition does not seem to be enough for a formal proof because it could occur that additional breakpoints in intermediate steps, which are canceled out later, also influence the behavior of the function at other places where we allow breakpoints in the end.

Another step towards resolving our conjecture may be to find an alternative proof of Theorem 2.5 not using Assumption 2.4. This might also be beneficial for generalizing our techniques to more hidden layers, since, while theoretically possible, a direct generalization is infeasible due to computational limitations.

In light of our results from Section 3, it would be desirable to provide a complete characterization of the functions contained in $\mathrm{ReLU}(k)$. Another potential research goal is improving our upper bounds on the width from Section 4 and/or proving matching lower bounds as discussed in Section 4.5.

Some more interesting research directions are the following: 1. Establishing or strengthening our results for special classes of NNs like recurrent neural networks (RNNs) or convolutional neural networks (CNNs), 2. Using exact representation results to show more drastic depth-width tradeoffs compared to existing results in the literature, 3. Understanding how the class $\mathrm{ReLU}(k)$ changes when a polynomial upper bound is imposed on the width of the NN; see related work by Vardi et al. [2021].

## Acknowledgments and Disclosure of Funding

Christoph Hertrich gratefully acknowledges funding by DFG-GRK 2434 "Facets of Complexity". Amitabh Basu gratefully acknowledges support from AFOSR Grant FA95502010341 and NSF Grant CCF2006587. Martin Skutella gratefully acknowledges funding by the Deutsche Forschungsgemeinschaft (DFG, German Research Foundation) under Germany's Excellence Strategy — The Berlin Mathematics Research Center MATH+ (EXC-2046/1, project ID: 390685689).

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
