# Appendix:

# Towards Lower Bounds on the Depth of ReLU Neural Networks

**Christoph Hertrich**, **Amitabh Basu**, **Marco Di Summa**, **Martin Skutella**

## A  Detailed Proof of Proposition 1.3

*Proof of Proposition 1.3.* It remains to argue that Conjecture 1.1 implies Conjecture 1.2. We prove the contraposition, that is, assuming that Conjecture 1.2 is violated, we show that Conjecture 1.1 is violated, as well. To this end, suppose there is a $k \in \mathbb{N}$, $n = 2^k$, such that $f_n$ is representable with $k$ hidden layers. We argue that under this hypothesis, any $(n + 1)$-term max function can be represented with $k$ hidden layers. To see this, observe that

$$\max\{\ell_1(x), \ldots, \ell_{n+1}(x)\} = \max\{0, \ell_1(x) - \ell_{n+1}(x), \ldots, \ell_n(x) - \ell_{n+1}(x)\} + \ell_{n+1}(x).$$

Modifying the first-layer weights of the NN computing $f_n$ such that input $x_i$ is replaced by the affine expression $\ell_i(x) - \ell_{n+1}(x)$, one obtains a $k$-hidden-layer NN computing the function $\max\{0, \ell_1(x) - \ell_{n+1}(x), \ldots, \ell_n(x) - \ell_{n+1}(x)\}$. Moreover, since affine functions, in particular also $\ell_{n+1}(x)$, can easily be represented by $k$-hidden-layer NNs, we obtain that any $(n + 1)$-term maximum is in $\mathrm{ReLU}_n(k)$. Using that any function in $\mathrm{CPWL}_n$ can be represented as linear combination of $(n + 1)$-term maximums [Wang and Sun, 2005], it follows that $\mathrm{ReLU}_n(k) = \mathrm{CPWL}_n$. In particular, since $k^* := \lceil \log_2(n + 1) \rceil = k + 1$, we obtain that Conjecture 1.1 must be violated as well. This concludes the proof of equivalence of Conjecture 1.1 and Conjecture 1.2. $\qquad\square$

## B  Detailed Proof of Theorem 2.5

Let us start with investigating the structure of the hyperplane arrangement $H$. For readers familiar with the interplay between hyperplane arrangements and polytopes, it is worth noting that $H$ is dual to a combinatorial equivalent of the 4-dimensional permutahedron. Hence, what we are studying in the following are some combinatorial properties of the permutahedron.

Recall that the regions of $H$ are given by the 120 polyhedra

$$C_\pi := \{x \in \mathbb{R}^4 \mid x_{\pi(0)} \leq x_{\pi(1)} \leq x_{\pi(2)} \leq x_{\pi(3)} \leq x_{\pi(4)}\}$$

for each permutation $\pi$ of $[4]_0$. With this representation, one can see that $C_\pi$ is a pointed polyhedral cone (with the origin as its only vertex) spanned by the four half-lines (a.k.a. *rays*)

$$R_{\{\pi(0)\}} := \{x \in \mathbb{R}^4 \mid x_{\pi(0)} \leq x_{\pi(1)} = x_{\pi(2)} = x_{\pi(3)} = x_{\pi(4)}\},$$
$$R_{\{\pi(0),\pi(1)\}} := \{x \in \mathbb{R}^4 \mid x_{\pi(0)} = x_{\pi(1)} \leq x_{\pi(2)} = x_{\pi(3)} = x_{\pi(4)}\},$$
$$R_{\{\pi(0),\pi(1),\pi(2)\}} := \{x \in \mathbb{R}^4 \mid x_{\pi(0)} = x_{\pi(1)} = x_{\pi(2)} \leq x_{\pi(3)} = x_{\pi(4)}\},$$
$$R_{\{\pi(0),\pi(1),\pi(2),\pi(3)\}} := \{x \in \mathbb{R}^4 \mid x_{\pi(0)} = x_{\pi(1)} = x_{\pi(2)} = x_{\pi(3)} \leq x_{\pi(4)}\}.$$

With that notation, we see that each of the 120 cells of $H$ is a simplicial cone spanned by four out of the 30 rays $R_S$ with $\emptyset \subsetneq S \subsetneq [4]_0$. For each such set $S$, denote its complement by $\bar{S} := [4]_0 \setminus S$. Let us use a generating vector $r_S \in \mathbb{R}^4$ for each of these rays such that $R_S = \mathrm{cone}\, r_S$ as follows: If $0 \in S$, then $r_S := \mathbb{1}_{\bar{S}} \in \mathbb{R}^4$, otherwise $r_S := -\mathbb{1}_S \in \mathbb{R}^4$, where for each $S \subseteq [4]$, the vector $\mathbb{1}_S \in \mathbb{R}^4$ contains entries 1 at precisely those index positions that are contained in $S$ and entries 0 elsewhere. For example, $r_{\{0,2,3\}} = (1, 0, 0, 1) \in \mathbb{R}^4$ and $r_{\{1,4\}} = (-1, 0, 0, -1) \in \mathbb{R}^4$. Then, the set $R$ containing conic generators of all the 30 rays of $H$ consists of the 30 vectors $R = (\{0, 1\}^4 \cup \{0, -1\}^4) \setminus \{0\}^4$.

Let $\mathcal{S}^{30}$ be the space of all $H$-conforming CPWL functions of type $\mathbb{R}^4 \to \mathbb{R}$. We show that $\mathcal{S}^{30}$ is a 30-dimensional vector space.

**Lemma B.1.** *The map $g \mapsto (g(r))_{r \in R}$ that evaluates a function $g \in \mathcal{S}^{30}$ at the 30 rays in $R$ is an isomorphism between $\mathcal{S}^{30}$ and $\mathbb{R}^{30}$. In particular, $\mathcal{S}^{30}$ is a 30-dimensional vector space.*

*Proof.* First note that $\mathcal{S}^{30}$ is closed under addition and scalar multiplication. Therefore, it is a subspace of the vector space of continuous functions of type $\mathbb{R}^4 \to \mathbb{R}$, and thus, in particular, a vector space. We show that the map $g \mapsto (g(r))_{r \in R}$ is in fact a vector space isomorphism. The map is obviously linear, so we only need to show that it is a bijection. In order to do so, remember that $\mathbb{R}^4$ is the union of the $5! = 120$ simplicial cones $C_\pi$. In particular, given the function values on the extreme rays of these cones, there is a unique positively homogeneous, continuous continuation that is linear within each of the 120 cones. This implies that the considered map is a bijection between $\mathcal{S}^{30}$ and $\mathbb{R}^{30}$. $\qquad\square$

The previous lemma also provides a canonical basis of the vector space $\mathcal{S}^{30}$: the one consisting of all CPWL functions attaining value 1 at one ray $r \in R$ and value 0 at all other rays. However, it turns out that for our purposes it is more convenient to work with a different basis. To this end, let $g_M(x) = \max_{i \in M} x_i$ for each $\{\emptyset, \{0\}\} \not\ni M \subseteq [4]_0$. These 30 functions contain, for example, the four (linear) coordinate projections $g_{\{i\}}(x) = x_i$, $i \in [4]$, and the function $f(x) = g_{[4]_0}(x) = \max\{0, x_1, x_2, x_3, x_4\}$.

**Lemma B.2.** *The 30 functions $g_M(x) = \max_{i \in M} x_i$ with $\{\emptyset, \{0\}\} \not\ni M \subseteq [4]_0$ form a basis of $\mathcal{S}^{30}$.*

*Proof.* Evaluating the 30 functions $g_M$ at all 30 rays $r \in R$ yields 30 vectors in $\mathbb{R}^{30}$. It can be easily verified (e.g., using a computer) that these vectors form a basis of $\mathbb{R}^{30}$. Thus, due to the isomorphism of Lemma B.1, the functions $g_M$ form a basis of $\mathcal{S}^{30}$. $\qquad\square$

Next, we focus on particular subspaces of $\mathcal{S}^{30}$ generated by only some of the 30 functions $g_M$. We prove that they correspond to the spaces of functions computable by $H$-conforming 2- and 3-layer NNs, respectively.

To this end, let $\mathcal{B}^{14}$ be the set of the 14 basis functions $g_M$ with $\{\emptyset, \{0\}\} \not\ni M \subseteq [4]_0$ and $|M| \leq 2$. Let $\mathcal{S}^{14}$ be the 14-dimensional subspace spanned by $\mathcal{B}^{14}$. Similarly, let $\mathcal{B}^{29}$ be the set of the 29 basis functions $g_M$ with $\{\emptyset, \{0\}\} \not\ni M \subsetneq [4]_0$ (all but $[4]_0$). Let $\mathcal{S}^{29}$ be the 29-dimensional subspace spanned by $\mathcal{B}^{29}$.

**Lemma B.3.** *The space $\mathcal{S}^{14}$ consists of all functions computable by $H$-conforming 2-layer NNs.*

*Proof.* Each function in $\mathcal{B}^{14}$ is a maximum of at most 2 numbers and can thus be represented by a 2-layer NN; compare Arora et al. [2018, Lemma D.3]. By putting the corresponding networks in parallel and adding appropriate weights to the connections to the output, also all linear combinations of these 14 functions, and thus, the full space $\mathcal{S}^{14}$, can be represented by a 2-layer NN.

Conversely, we show that any function representable by a 2-layer NN is indeed contained in $\mathcal{S}^{14}$. It suffices to show that the output of every neuron in the first (and only) hidden layer of an $H$-conforming ReLU NN is in $\mathcal{S}^{14}$ because the output of a 2-layer NN is a linear combination of such outputs. Let $a \in \mathbb{R}^4$ be the first-layer weights of such a neuron, computing the function $g_a(x) := \max\{a^T x, 0\}$, which has the hyperplane $\{x \in \mathbb{R}^4 \mid a^T x = 0\}$ as breakpoints (or is constantly zero). Since the NN must be $H$-conforming, this must be one of the ten hyperplanes $x_i = x_j$, $0 \leq i < j \leq 4$. Thus, $g_a(x) = \max\{\lambda(x_i - x_j), 0\}$ for some $\lambda \in \mathbb{R}$. If $\lambda \geq 0$, it follows that $g_a = \lambda g_{\{i,j\}} - \lambda g_{\{j\}} \in \mathcal{S}^{14}$, and if $\lambda \leq 0$, we obtain $g_a = -\lambda g_{\{i,j\}} + \lambda g_{\{i\}} \in \mathcal{S}^{14}$. This concludes the proof. $\qquad\square$

For 3-layer NNs, an analogous statement can be made. However, only one direction can be easily seen.

**Lemma B.4.** *Any function in $\mathcal{S}^{29}$ can be represented by an $H$-conforming 3-layer NN.*

*Proof.* Each function in $\mathcal{B}^{29}$ is a maximum of at most 4 numbers and can thus be represented by a 3-layer NN; compare Arora et al. [2018, Lemma D.3]. As in the previous lemma, also linear combinations of those can be represented. $\qquad\square$

Our goal is to prove the converse as well: any $H$-conforming function represented by a 3-layer NN is in $\mathcal{S}^{29}$. Since $f(x) = \max\{0, x_1, x_2, x_3, x_4\}$ is the 30-th basis function, which is linearly independent from $\mathcal{B}^{29}$ and thus not contained in $\mathcal{S}^{29}$, this implies Theorem 2.5. To achieve this goal, we first provide another characterization of $\mathcal{S}^{29}$, which can be seen as an orthogonal direction to $\mathcal{S}^{29}$ in $\mathcal{S}^{30}$. For a function $g \in \mathcal{S}^{30}$, let

$$\phi(g) := \sum_{\emptyset \subsetneq S \subsetneq [4]_0} (-1)^{|S|} g(r_S)$$

be a linear map from $\mathcal{S}^{30}$ to $\mathbb{R}$.

**Lemma B.5.** *A function $g \in \mathcal{S}^{30}$ is contained in $\mathcal{S}^{29}$ if and only if $\phi(g) = 0$.*

*Proof.* Any $g \in \mathcal{S}^{30}$ can be represented as a unique linear combination of the 30 basis functions $g_M$ and is contained in $\mathcal{S}^{29}$ if and only if the coefficient of $f = g_{[4]_0}$ is zero. One can easily check (with a computer) that all functions in $\mathcal{B}^{29}$ are mapped to 0 by $\phi$, but the 30-th basis function $f$ is not. Thus, $g$ is contained in $\mathcal{S}^{29}$ if and only if it satisfies $\phi(g) = 0$. $\qquad\square$

In order to make use of our Assumption 2.4, we need the following insight about when the property of being $H$-conforming is preserved after applying a ReLU activation.

**Lemma B.6.** *Let $g \in \mathcal{S}^{30}$. The function $h = \sigma \circ g$ is $H$-conforming (and thus in $\mathcal{S}^{30}$ as well) if and only if there is no pair of sets $\emptyset \subsetneq S \subsetneq S' \subsetneq [4]_0$ with $g(r_S)$ and $g(r_{S'})$ being nonzero and having different signs.*

*Proof.* The key observation to prove this lemma is the following: for two rays $r_S$ and $r_{S'}$, there exists a cell $C$ of the hyperplane arrangement $H$ for which both $r_S$ and $r_{S'}$ are extreme rays if and only if $S \subsetneq S'$ or $S' \subsetneq S$.

Hence, if there exists a pair of sets $\emptyset \subsetneq S \subsetneq S' \subsetneq [4]_0$ with $g(r_S)$ and $g(r_{S'})$ being nonzero and having different signs, then the function $g$ restricted to $C$ is a linear function with both strictly positive and strictly negative values. Therefore, after applying the ReLU activation, the resulting function $h$ has breakpoints within $C$ and is not $H$-conforming.

Conversely, if for each pair of sets $\emptyset \subsetneq S \subsetneq S' \subsetneq [4]_0$, both $g(r_S)$ and $g(r_{S'})$ are either nonpositive or nonnegative, then $g$ restricted to any cell $C$ of $H$ is either nonpositive or nonnegative everywhere. In the first case, $h$ restricted to that cell $C$ is the zero function, while in the second case, $h$ coincides with $g$ in $C$. In both cases, $h$ is linear within all cells and, thus, $H$-conforming. $\qquad\square$

Having collected all these lemmas, we are finally able to construct a MIP whose solution proves that any function computed by an $H$-conforming 3-layer NN is in $\mathcal{S}^{29}$. As in the proof of Lemma B.3, it suffices to focus on the output of a single neuron in the second hidden layer. Let $h = \sigma \circ g$ be the output of such a neuron with $g$ being its input. Observe that $g$ is by construction a function computed by a 2-layer NN, and thus, by Lemma B.3, a linear combination of the 14 functions in $\mathcal{B}^{14}$. The MIP contains three types of variables, which we denote in bold to distinguish them from constants:

- 14 continuous variables $\mathbf{a}_M \in [-1, 1]$, being the coefficients of the linear combination of the basis of $\mathcal{S}^{14}$ forming $g$, that is, $g = \sum_{g_M \in \mathcal{B}^{14}} \mathbf{a}_M g_M$ (since multiplying $g$ and $h$ with a nonzero scalar does not alter containment of $h$ in $\mathcal{S}^{29}$, we may assume unit interval bounds),

- 30 binary variables $\mathbf{z}_S \in \{0, 1\}$ for $\emptyset \subsetneq S \subsetneq [4]_0$, determining whether the considered neuron is strictly active at ray $r_S$, that is, whether $g(r_S) > 0$,

- 30 continuous variables $\mathbf{y}_S \in \mathbb{R}$ for $\emptyset \subsetneq S \subsetneq [4]_0$, representing the output of the considered neuron at all rays, that is, $\mathbf{y}_S = h(r_S)$.

To ensure that these variables interact as expected, we need two types of constraints:

- For each of the 30 rays $r_S$, $\emptyset \subsetneq S \subsetneq [4]_0$, the following constraints ensure that $\mathbf{z}_S$ and output $\mathbf{y}_S$ are correctly calculated from the variables $\mathbf{a}_M$, that is, $\mathbf{z}_S = 1$ if and only if $g(r_S) = \sum_{g_M \in \mathcal{B}^{14}} \mathbf{a}_M g_M(r_S)$ is positive, and $\mathbf{y}_S = \max\{0, g(r_S)\}$. Also compare the references given in the introduction concerning MIP models for ReLU units. Note that

the restriction of the coefficients $\mathbf{a}_M$ to $[-1, 1]$ ensures that the absolute value of $g(r_S)$ is always bounded by 14, allowing us to use 15 as a replacement for $+\infty$:

$$\begin{aligned}
\mathbf{y}_S &\geq 0 \\
\mathbf{y}_S &\geq \sum_{g_M \in \mathcal{B}^{14}} \mathbf{a}_M g_M(r_S) \\
\mathbf{y}_S &\leq 15 \mathbf{z}_S \\
\mathbf{y}_S &\leq \sum_{g_M \in \mathcal{B}^{14}} \mathbf{a}_M g_M(r_S) + 15(1 - \mathbf{z}_S)
\end{aligned} \tag{3}$$

Observe that these constraints ensure that one of the following two cases happens: if $\mathbf{z}_S = 0$, then the first and third line imply $\mathbf{y}_S = 0$ and the second line implies that the incoming activation is in fact nonpositive. The fourth line is always satisfied in that case. Otherwise, if $\mathbf{z}_S = 1$, then the second and fourth line imply that $\mathbf{y}_S$ equals the incoming activation, and, in combination with the first line, this has to be nonnegative. The third line is always satisfied in that case. Hence, the set of constraints (3) correctly models the ReLU activation function.

- For each of the 150 pairs of sets $\emptyset \subsetneq S \subsetneq S' \subsetneq [4]_0$, the following constraints ensure that the property in Lemma B.6 is satisfied. More precisely, if one of the variables $\mathbf{z}_S$ or $\mathbf{z}_{S'}$ equals 1, then the ray of the other set has nonnegative activation, that is, $g(r_{S'}) \geq 0$ or $g(r_S) \geq 0$, respectively:

$$\begin{aligned}
\sum_{g_M \in \mathcal{B}^{14}} \mathbf{a}_M g_M(r_S) &\geq 15(\mathbf{z}_{S'} - 1) \\
\sum_{g_M \in \mathcal{B}^{14}} \mathbf{a}_M g_M(r_{S'}) &\geq 15(\mathbf{z}_S - 1)
\end{aligned} \tag{4}$$

Observe that these constraints successfully prevent that the two rays $r_S$ and $r_{S'}$ have nonzero activations with different signs. Conversely, if this is not the case, then we can always satisfy constraints (4) by setting only those variables $\mathbf{z}_S$ to value 1 where the activation of ray $r_S$ is *strictly* positive. (Note that, if the incoming activation is precisely zero, constraints (3) make it possible to choose both values 0 or 1 for $\mathbf{z}_S$.) Hence, these constraints are in fact appropriate to model $H$-conformity.

In the light of Lemma B.5, the objective function of our MIP is to maximize $\phi(h)$, that is, the expression

$$\sum_{\emptyset \subsetneq S \subsetneq [4]_0} (-1)^{|S|} \mathbf{y}_S.$$

The MIP has a total of 30 binary and 44 continuous variables, as well as 420 inequality constraints. The next proposition formalizes how this MIP can be used to check whether a 3-layer NN function can exist outside $\mathcal{S}^{29}$.

**Proposition B.7.** *There exists an $H$-conforming 3-layer NN computing a function not contained in $\mathcal{S}^{29}$ if and only if the objective value of the MIP defined above is strictly positive.*

*Proof.* For the first direction, assume that such an NN exists. Since its final output is a linear combination of the outputs of the neurons in the second hidden layer, one of these neurons must compute a function $\tilde{h} = \sigma \circ \tilde{g} \notin \mathcal{S}^{29}$, with $\tilde{g}$ being the input to that neuron. By Lemma B.5, it follows that $\phi(\tilde{h}) \neq 0$. Moreover, we can even assume without loss of generality that $\phi(\tilde{h}) > 0$, as we argue now. If this is not the case, multiply all first-layer weights of the NN by $-1$ to obtain a new NN computing function $\hat{h}$ instead of $\tilde{h}$. Observing that $r_S = -r_{[4]_0 \setminus S}$ for all $r_S \in R$, we obtain $\hat{h}(r_S) = \tilde{h}(-r_S) = \tilde{h}(r_{[4]_0 \setminus S})$ for all $r_S \in R$. Plugging this into the definition of $\phi$ and using that the cardinalities of $S$ and $[4]_0 \setminus S$ have different parity, we further obtain $\phi(\hat{h}) = -\phi(\tilde{h})$. Therefore, we can assume that $\phi(\tilde{h})$ was already positive in the first place.

Using Lemma B.3, $\tilde{g}$ can be represented as a linear combination $\tilde{g} = \sum_{g_M \in \mathcal{B}^{14}} \tilde{\mathbf{a}}_M g_M$ of the functions in $\mathcal{B}^{14}$. Let $\alpha := \max_M |\tilde{\mathbf{a}}_M| > 0$. Let us define modified functions $g$ and $h$ from $\tilde{g}$ and $\tilde{h}$

as follows. Let $\mathbf{a}_M := \tilde{\mathbf{a}}_M/\alpha \in [-1, 1]$, $g := \sum_{g_M \in \mathcal{B}^{14}} \mathbf{a}_M g_M$, and $h := \sigma \circ g$. Moreover, for all rays $r_S \in R$, let $\mathbf{y}_S := h(r_S)$, as well as $\mathbf{z}_S := 1$ if $\mathbf{y}_S > 0$, and $\mathbf{z}_S := 0$ otherwise.

It is easy to verify that the variables $\mathbf{a}_M$, $\mathbf{y}_S$, and $\mathbf{z}_S$ defined that way satisfy (3). Moreover, since the NN is $H$-conforming, they also satisfy (4). Finally, they also yield a strictly positive objective function value since $\phi(h) = \phi(\tilde{h})/\alpha > 0$.

For the reverse direction, assume that there exists a MIP solution consisting of $\mathbf{a}_M$, $\mathbf{y}_S$, and $\mathbf{z}_S$, satisfying (3) and (4), and having a strictly positive objective function value. Define the functions $g := \sum_{g_M \in \mathcal{B}^{14}} \mathbf{a}_M g_M$ and $h := \sigma \circ g$. One concludes from (3) that $h(r_S) = \mathbf{y}_S$ for all rays $r_S \in R$. Lemma B.3 implies that $g$ can be represented by a 2-layer NN. Thus, $h$ can be represented by a 3-layer NN. Moreover, constraints (4) guarantee that this NN is $H$-conforming. Finally, since the MIP solution has strictly positive objective function value, we obtain $\phi(h) > 0$, implying that $h \notin \mathcal{S}^{29}$. $\square$

In order to use the MIP as part of a mathematical proof, we employed a MIP solver that uses exact rational arithmetics without numerical errors, namely the solver by the Parma Polyhedral Library (PPL) [Bagnara et al., 2008]. We called the solver from a SageMath [The Sage Developers, 2020] script on a machine with an Intel Core i7-8700 6-Core 64-bit CPU and 15.5 GB RAM, using the openSUSE Leap 15.2 Linux distribution. SageMath, which natively includes the PPL solver, is published under the GPLv3 license. After a total running time of almost 7 days (153 hours), we obtained optimal objective function value zero. This makes it possible to prove Theorem 2.5.

*Proof of Theorem 2.5.* Since the MIP has optimal objective function value zero, Proposition B.7 implies that any function computed by an $H$-conforming 3-layer NN is contained in $\mathcal{S}^{29}$. In particular, under Assumption 2.4, the function $f(x) = \max\{0, x_1, x_2, x_3, x_4\}$ cannot be computed by a 3-layer NN. $\square$

We remark that state-of-the-art MIP solver Gurobi (version 9.1.1) [Gurobi Optimization, LLC, 2021], which is commercial but offers free academic licenses, is able to solve the same MIP within less than a second, providing the same result. However, Gurobi does not employ exact arithmetics, making it impossible to exclude numerical errors and use it as a mathematical proof.

The SageMath code can be found on GitHub.[3] Additionally, the MIP can be found there as `.mps` file, a standard format to represent MIPs. This allows one to use any solver of choice to reproduce our result.

## C   Detailed Proof of Proposition 3.3

Since the proof exploits some properties of the underlying polyhedral complex of the considered CPWL functions, we will first introduce some terminology, notation and results related to polyhedral complexes in $\mathbb{R}^n$ for any $n \geq 1$.

**Definition C.1.** Given an abelian group $(G, +)$, we define $\mathcal{F}^n(G)$ as the family of all functions $\phi$ of type $\phi \colon \mathcal{P}^n \to G$, where $\mathcal{P}$ is a polyhedral complex that covers $\mathbb{R}^n$. We say that $\mathcal{P}$ is the *underlying* polyhedral complex, or the polyhedral complex *associated* with $\phi$.

Just to give an intuition of the reason for this definition, let us mention that later we will choose $(G, +)$ to be the set of affine linear maps $\mathbb{R}^n \to \mathbb{R}$ with respect to the standard operation of sum of functions. Moreover, given a convex CPWL function $f \colon \mathbb{R}^n \to \mathbb{R}$ with underlying polyhedral complex $\mathcal{P}$, we will consider the following function $\phi \in \mathcal{F}^n(G)$: for every $P \in \mathcal{P}^n$, $\phi(P)$ will be the affine linear map that coincides with $f$ over $P$. It can be useful, though not necessary, to keep this in mind when reading the next definitions and observations.

It is useful to observe that the functions in $\mathcal{F}^n(G)$ can also be described in a different way. Before explaining this, we need to define an ordering between the two elements of each pair of opposite halfspaces. More precisely, let $H$ be a hyperplane in $\mathbb{R}^n$ and let $H'$, $H''$ be the two closed halfspaces delimited by $H$. We choose an arbitrary rule to say that $H'$ "precedes" $H''$, which we write as

---

[3]`https://github.com/ChristophHertrich/relu-mip-depth-bound`

$H' \prec H''$.[4] We can then extend this ordering rule to some pairs of $n$-dimensional polyhedra of a polyhedral complex in $\mathbb{R}^n$. Specifically, given a polyhedral complex $\mathcal{P}$ in $\mathbb{R}^n$, let $P', P'' \in \mathcal{P}^n$ be such that $F := P' \cap P'' \in \mathcal{P}^{n-1}$. Further, let $H$ be the unique hyperplane containing $F$. We say that $P' \prec P''$ if the halfspace delimited by $H$ and containing $P'$ precedes the halfspace delimited by $H$ and containing $P''$.

We can now explain the alternate description of the functions in $\mathcal{F}^n(G)$, which is based on the following notion.

**Definition C.2.** Let $\phi \in \mathcal{F}^n(G)$, with associated polyhedral complex $\mathcal{P}$. The *facet-function* associated with $\phi$ is the function $\psi \colon \mathcal{P}^{n-1} \to G$ defined as follows: given $F \in \mathcal{P}^{n-1}$, let $P', P''$ be the two polyhedra in $\mathcal{P}^n$ such that $F = P' \cap P''$, where $P' \prec P''$; then we set $\psi(F) := \phi(P') - \phi(P'')$.

Although it will not be used, we observe that knowing $\psi$ is sufficient to reconstruct $\phi$ up to an additive constant. This means that a function $\phi' \in \mathcal{F}^n(G)$ associated with the same polyhedral complex $\mathcal{P}$ has the same facet-function $\psi$ if and only if there exists $g \in G$ such that $\phi(P) - \phi'(P) = g$ for every $P \in \mathcal{P}^n$. (However, it is not true that every function $\psi \colon \mathcal{P}^{n-1} \to G$ is the facet-function of some function in $\mathcal{F}^n(G)$.)

We now introduce a sum operation over $\mathcal{F}^n(G)$.

**Definition C.3.** Given $p$ functions $\phi_1, \ldots, \phi_p \in \mathcal{F}^n(G)$ with associated polyhedral complexes $\mathcal{P}_1, \ldots, \mathcal{P}_p$, the sum $\phi := \phi_1 + \cdots + \phi_p$ is the function in $\mathcal{F}^n(G)$ defined as follows:

- the polyhedral complex associated with $\phi$ is $\mathcal{P} := \{P_1 \cap \cdots \cap P_p \mid P_i \in \mathcal{P}_i \text{ for every } i\}$;

- given $P \in \mathcal{P}^n$, $P$ can be uniquely obtained as $P_1 \cap \cdots \cap P_p$, where $P_i \in \mathcal{P}_i^n$ for every $i$; we then define

$$\phi(P) = \sum_{i=1}^{p} \phi_i(P_i).$$

The term "sum" is justified by the fact that when $\mathcal{P}_1 = \cdots = \mathcal{P}_p$ (and thus $\phi_1, \ldots, \phi_p$ have the same domain) we obtain the standard notion of sum of functions.

The next results shows how to compute the facet-function of a sum of functions in $\mathcal{F}^n(G)$.

**Observation C.4.** *With the notation of Definition C.3, let $\psi_1, \ldots, \psi_p$ be the facet-functions associated with $\phi_1, \ldots, \phi_p$, and let $\psi$ be the facet-function associated with $\phi$. Given $F \in \mathcal{P}^{n-1}$, let $I$ be the set of indices $i \in \{1, \ldots, p\}$ such that $\mathcal{P}_i^{n-1}$ contains a (unique) element $F_i$ with $F \subseteq F_i$. Then*

$$\psi(F) = \sum_{i \in I} \psi_i(F_i). \tag{5}$$

*Proof.* Let $P', P''$ be the two polyhedra in $\mathcal{P}^n$ such that $F = P' \cap P''$, with $P' \prec P''$. We have $P' = P'_1 \cap \cdots \cap P'_p$ and $P'' = P''_1 \cap \cdots \cap P''_p$ for a unique choice of $P'_i, P''_i \in \mathcal{P}_i^n$ for every $i$. Then

$$\psi(F) = \phi(P') - \phi(P'') = \sum_{i=1}^{p} (\phi_i(P'_i) - \phi_i(P''_i)). \tag{6}$$

Now fix $i \in [p]$. Since $F \subseteq P'_i \cap P''_i$, $\dim(P'_i \cap P''_i) \geq n - 1$. If $\dim(P'_i \cap P''_i) = n - 1$, then $F_i := P'_i \cap P''_i \in \mathcal{P}_i^{n-1}$ and $\phi_i(P'_i) - \phi_i(P''_i) = \psi_i(F_i)$. Furthermore, $i \in I$ because $F \subseteq F_i$. If, on the contrary, $\dim(P'_i \cap P''_i) = n$, the fact that $\mathcal{P}_i$ is a polyhedral complex implies that $P'_i = P''_i$, and thus $\phi_i(P'_i) - \phi_i(P''_i) = 0$. Moreover, in this case $i \notin I$: this is because $P' \cup P'' \subseteq P'_i$, which implies that the relative interior of $F$ is contained in the relative interior of $P'_i$. With these observations, from (6) we obtain (5). $\square$

**Definition C.5.** Fix $\phi \in \mathcal{F}^n(G)$, with associated polyhedral complex $\mathcal{P}$. Let $H$ be a hyperplane in $\mathbb{R}^n$, and let $H', H''$ be the closed halfspaces delimited by $H$. Define the polyhedral complex

$$\widehat{\mathcal{P}} = \{P \cap H \mid P \in \mathcal{P}\} \cup \{P \cap H' \mid P \in \mathcal{P}\} \cup \{P \cap H'' \mid P \in \mathcal{P}\}.$$

---

[4]In case one wants to see such a rule explicitly, this is one way. Fix an arbitrary $\bar{x} \in H$. We can say that $H' \prec H''$ if and only if $\bar{x} + e_i \in H'$, where $e_i$ is the first vector in the standard basis of $\mathbb{R}^d$ that does not lie on $H$ (i.e., $e_1, \ldots, e_{i-1} \in H$ and $e_i \notin H$). Note that this definition does not depend on the choice of $\bar{x}$.

The refinement of $\phi$ with respect to $H$ is the function $\widehat{\phi} \in \mathcal{F}^n(G)$ with associated polyhedral complex $\widehat{\mathcal{P}}$ defined as follows: given $\widehat{P} \in \widehat{\mathcal{P}}^n$, $\widehat{\phi}(\widehat{P}) := \phi(P)$, where $P$ is the unique polyhedron in $\mathcal{P}$ that contains $\widehat{P}$.

The next results shows how to compute the facet-function of a refinement.

**Observation C.6.** *With the notation of Definition C.5, let $\psi$ be the facet-function associated with $\phi$. Then the facet-function $\widehat{\psi}$ associated with $\widehat{\phi}$ is as follows: for every $\widehat{F} \in \widehat{\mathcal{P}}^{n-1}$,*

$$\widehat{\psi}(\widehat{F}) = \begin{cases} \psi(F) & \text{if there exists a (unique) } F \in \mathcal{P}^{n-1} \text{ containing } \widehat{F} \\ 0 & \text{otherwise.} \end{cases}$$

*Proof.* Let $\widehat{P}', \widehat{P}''$ be the polyhedra in $\widehat{\mathcal{P}}^n$ such that $\widehat{F} = \widehat{P}' \cap \widehat{P}''$, with $\widehat{P}' \prec \widehat{P}''$. Further, let $P', P''$ be the unique polyhedra in $\mathcal{P}^n$ that contain $\widehat{P}', \widehat{P}''$ (respectively); note that $P' \prec P''$.

If there exists $F \in \mathcal{P}^{n-1}$ containing $\widehat{F}$, then the fact that $\mathcal{P}$ is a polyhedral complex implies that $F = P' \cap P''$. Thus $\widehat{\psi}(\widehat{F}) = \widehat{\phi}(\widehat{P}') - \widehat{\phi}(\widehat{P}'') = \phi(P') - \phi(P'') = \psi(F)$.

Assume now that no element of $\mathcal{P}^{n-1}$ contains $\widehat{F}$. Then there exists $P \in \mathcal{P}^n$ such that $\widehat{F} = P \cap H$ and $H$ intersects the interior of $P$. Then $\widehat{P}' = P \cap H'$ and $\widehat{P}'' = P \cap H''$ (or vice versa). It follows that $\widehat{\psi}(\widehat{F}) = \widehat{\phi}(\widehat{P}') - \widehat{\phi}(\widehat{P}'') = \phi(P) - \phi(P) = 0$. $\qquad\square$

We now prove that the operations of sum and refinement commute: the refinement of a sum is the sum of the refinements.

**Observation C.7.** *Let $p$ functions $\phi_1, \ldots, \phi_p \in \mathcal{F}^n(G)$, with associated polyhedral complexes $\mathcal{P}_1, \ldots, \mathcal{P}_p$, be given. Define $\phi := \phi_1 + \cdots + \phi_p$. Let $H$ be a hyperplane in $\mathbb{R}^n$, and let $H', H''$ be the closed halfspaces delimited by $H$. Then $\widehat{\phi} = \widehat{\phi}_1 + \cdots + \widehat{\phi}_p$.*

*Proof.* Define $\widetilde{\phi} := \widehat{\phi}_1 + \cdots + \widehat{\phi}_p$. It can be verified that $\widehat{\phi}$ and $\widetilde{\phi}$ are defined on the same polyhedral complex, which we denote by $\widehat{\mathcal{P}}$. We now fix $\widehat{P} \in \widehat{\mathcal{P}}^n$ and show that $\widehat{\phi}(\widehat{P}) = \widetilde{\phi}(\widehat{P})$.

Since $\widehat{P} \in \widehat{\mathcal{P}}^n$, we have $\widehat{P} = P_1 \cap \cdots \cap P_p \cap H'$, where $P_i \in \mathcal{P}_i^n$ for every $i$. (We ignore the case $\widehat{P} = P_1 \cap \cdots \cap P_p \cap H''$, which is identical.) Then

$$\widehat{\phi}(\widehat{P}) = \phi(P_1 \cap \cdots \cap P_p) = \sum_{i=1}^{p} \phi_i(P_i) = \sum_{i=1}^{p} \widehat{\phi}_i(P_i \cap H') = \widetilde{\phi}(P_1 \cap \cdots \cap P_p \cap H') = \widetilde{\phi}(\widehat{P}),$$

where the first and third equations follow from the definition of refinement, while the second and fourth equations follow from the definition of sum. $\qquad\square$

The *lineality space* of a (nonempty) polyhedron $P = \{x \in \mathbb{R}^n \mid Ax \leq b\}$ is the null space of the constraint matrix $A$. In other words, it is the set of vectors $y \in \mathbb{R}^n$ such that for every $x \in P$ the whole line $\{x + \lambda y \mid \lambda \in \mathbb{R}\}$ is a subset of $P$. We say that the lineality space of $P$ is *trivial*, if it contains only the zero vector, and *nontrivial* otherwise.

Since, given a polyhedral complex $\mathcal{P}$ that covers $\mathbb{R}^n$, all the nonempty polyhedra in $\mathcal{P}$ share the same lineality space $L$, we will call $L$ the lineality space of $\mathcal{P}$.

**Lemma C.8.** *Given an abelian group $(G, +)$, pick $\phi_1, \ldots, \phi_p \in \mathcal{F}^n(G)$, with associated polyhedral complexes $\mathcal{P}_1, \ldots, \mathcal{P}_p$. Assume that for every $i \in [p]$ the lineality space of $\mathcal{P}_i$ is nontrivial. Define $\phi := \phi_1 + \cdots + \phi_p$, $\mathcal{P}$ as the underlying polyhedral complex, and $\psi$ as the facet-function of $\phi$. Then for every hyperplane $H \subseteq \mathbb{R}^n$, the set*

$$S := \bigcup \{F \in \mathcal{P}^{n-1} \mid F \subseteq H, \psi(F) \neq 0\}$$

*is either empty or contains a line.*

*Proof.* The proof is by induction on $n$. For $n = 1$, the assumptions imply that all $\mathcal{P}_i$ are equal to $\mathcal{P}$, and each of these polyhedral complexes has $\mathbb{R}$ as its only nonempty face. Since $\mathcal{P}^{n-1}$ is empty, no hyperplane $H$ such that $S \neq \emptyset$ can exist.

Now fix $n \geq 2$. Assume by contradiction that there exists a hyperplane $H$ such that $S$ is nonempty and contains no line. Let $\widehat{\phi}$ be the refinement of $\phi$ with respect to $H$, $\widehat{\mathcal{P}}$ be the underlying polyhedral complex, and $\widehat{\psi}$ be the associated facet-function. Further, we define $\mathcal{Q} := \{P \cap H \mid P \in \widehat{\mathcal{P}}\}$, which is a polyhedral complex that covers $H$. Note that if $H$ is identified with $\mathbb{R}^{n-1}$ then we can think of $\mathcal{Q}$ as a polyhedral complex that covers $\mathbb{R}^{n-1}$, and the restriction of $\widehat{\psi}$ to $\mathcal{Q}^{n-1}$, which we denote by $\phi'$, can be seen as a function in $\mathcal{F}^{n-1}(G)$. We will prove that $\phi'$ does not satisfy the lemma, contradicting the inductive hypothesis.

Since $\phi = \phi_1 + \cdots + \phi_p$, by Observation C.7 we have $\widehat{\phi} = \widehat{\phi}_1 + \cdots + \widehat{\phi}_p$. Note that for every $i \in [p]$ the hyperplane $H$ is covered by the elements of $\widehat{\mathcal{P}}^{n-1}$. This implies that for every $\widehat{F} \in \widehat{\mathcal{P}}^{n-1}$ and $i \in [p]$ there exists $\widehat{F}_i \in \widehat{\mathcal{P}}_i^{n-1}$ such that $\widehat{F} \subseteq \widehat{F}_i$. Then, by Observation C.4, $\widehat{\psi}(\widehat{F}) = \widehat{\psi}_1(\widehat{F}_1) + \cdots + \widehat{\psi}_p(\widehat{F}_p)$.

Now, additionally suppose that $\widehat{F}$ is contained in $H$, that is, $\widehat{F} \in \mathcal{Q}^{n-1}$. Let $i \in [p]$ be such that the lineality space of $\mathcal{P}_i$ is not parallel to $H$. Then no element of $\mathcal{P}_i^{n-1}$ contains $\widehat{F}_i$. By Observation C.6, $\widehat{\psi}_i(\widehat{F}_i) = 0$. We then conclude that

$$\widehat{\psi}(\widehat{F}) = \sum_{i \in J} \widehat{\psi}_i(\widehat{F}_i) \quad \text{for every } \widehat{F} \in \mathcal{Q}^{n-1},$$

where $J$ is the set of indices $i$ such that the lineality space of $\mathcal{P}_i$ is parallel to $H$. This means that

$$\phi' = \sum_{i \in J} \phi_i',$$

where $\phi_i'$ is the restriction of $\widehat{\psi}_i$ to $\mathcal{Q}_i^{n-1}$, with $\mathcal{Q}_i := \{P \cap H \mid P \in \widehat{\mathcal{P}}_i\}$. Note that for every $i \in J$ the lineality space of $\mathcal{Q}_i$ is clearly nontrivial, as it coincides with the lineality space of $\mathcal{P}_i$.

Now pick any $\widehat{F} \in \mathcal{Q}^{n-1}$. Note that if there exists $F \in \mathcal{P}^{n-1}$ such that $\widehat{F} \subseteq F$, then $\widehat{F} = F$. It then follows from Observation C.6 that

$$\bigcup \left\{ \widehat{F} \in \mathcal{Q}^{n-1} \,\middle|\, \widehat{\psi}(\widehat{F}) \neq 0 \right\} = S.$$

In other words,

$$\bigcup \left\{ F \in \mathcal{Q}^{n-1} \,\middle|\, \phi'(F) \neq 0 \right\} = S. \tag{7}$$

Since $S \neq H$ (as $S$ contains no line), there exists a polyhedron $F \in \mathcal{Q}^{n-1}$ such that $F \subseteq S$ and $F$ has a facet $F_0$ which does not belong to any other polyhedron in $\mathcal{Q}^{n-1}$ contained in $S$. Then the facet-function $\psi'$ associated with $\phi'$ satisfies $\psi'(F_0) \neq 0$. Let $H'$ be the $(n-2)$-dimensional affine space containing $F_0$. Then the set

$$S' := \bigcup \left\{ F \in \mathcal{Q}^{n-2} \,\middle|\, F \subseteq H',\, \psi'(F) \neq 0 \right\}$$

is nonempty, as $F_0 \subseteq S'$. Furthermore, we claim that $S'$ contains no line. To see why this is true, take any $F \in \mathcal{Q}^{n-2}$ such that $F \subseteq H'$ and $\psi'(F) \neq 0$, and let $F', F''$ be the two polyhedra in $\mathcal{Q}^{n-1}$ having $F$ as facet. Then $\phi'(F') \neq \phi'(F'')$, and thus at least one of these values (say $\phi'(F')$) is nonzero. Then, by (7), $F' \subseteq S$, and thus also $F \subseteq S$. This shows that $S' \subseteq S$ and therefore $S'$ contains no line.

We have shown that $\phi'$ does not satisfy the lemma. This contradicts the inductive assumption that the lemma holds in dimension $n - 1$. □

Finally, we can use this lemma to prove Proposition 3.3.

*Proof of Proposition 3.3.* Assume for the sake of a contradiction that

$$f_0(x) = \sum_{i=1}^{p} \lambda_i \max\{\ell_{i1}(x), \ldots, \ell_{in}(x)\} \quad \text{for every } x \in \mathbb{R}^n,$$

where $p \in \mathbb{N}$, $\lambda_1, \ldots, \lambda_p \in \mathbb{R}$ and $\ell_{ij} \colon \mathbb{R}^n \to \mathbb{R}$ is an affine linear function for every $i \in [p]$ and $j \in [n]$. Define $f_i(x) := \lambda_i \max\{\ell_{i1}(x), \ldots, \ell_{in}(x)\}$ for every $i \in [p]$, which is a CPWL function.

Fix any $i \in [p]$ such that $\lambda_i \geq 0$. Then $f_i$ is convex. Note that its epigraph $E_i := \{(x, z) \in \mathbb{R}^n \times \mathbb{R} \mid z \geq \ell_{ij}(x) \text{ for } j \in [n]\}$ is a polyhedron in $\mathbb{R}^{n+1}$ defined by $n$ inequalities, and thus has nontrivial lineality space. Furthermore, the line orthogonal to the $x$-space is not contained in $E_i$. Since the underlying polyhedral complex $\mathcal{P}_i$ of $f_i$ consists of the orthogonal projections of the faces of $E_i$ (excluding $E_i$ itself) onto the $x$-space, this implies that $\mathcal{P}_i$ has also nontrivial lineality space. (More precisely, the lineality space of $\mathcal{P}_i$ is the projection of the lineality space of $E_i$.)

If $\lambda_i < 0$, then $f_i$ is concave. By arguing as above on the convex function $-f_i$, one obtains that the underlying polyhedral complex $\mathcal{P}_i$ has again nontrivial lineality space. Thus this property holds for every $i \in [p]$.

The set of affine linear functions $\mathbb{R}^n \to \mathbb{R}$ forms an abelian group (with respect to the standard operation of sum of functions), which we denote by $(G, +)$. For every $i \in [p]_0$, let $\phi_i$ be the function in $\mathcal{F}^n(G)$ with underlying polyhedral complex $\mathcal{P}_i$ defined as follows: for every $P \in \mathcal{P}_i^n$, $\phi_i(P)$ is the affine linear function that coincides with $f_i$ over $P$. Define $\phi := \phi_1 + \cdots + \phi_p$ and let $\mathcal{P}$ be the underlying polyhedral complex.

Note that for every $P \in \mathcal{P}^n$, $\phi(P)$ is precisely the affine linear function that coincides with $f_0$ within $P$. However, $\mathcal{P}$ may not coincide with $\mathcal{P}_0$, as there might exist $P', P'' \in \mathcal{P}^d$ sharing a facet such that $\phi(P') = \phi(P'')$; when this happens, $f_0$ is affine linear over $P' \cup P''$ and therefore $P'$ and $P''$ are merged together in $\mathcal{P}_0$. Nonetheless, $\mathcal{P}$ is a refinement of $\mathcal{P}_0$, i.e., for every $P \in \mathcal{P}_0^n$ there exist $P_1, \ldots, P_k \in \mathcal{P}^n$ (for some $k \geq 1$) such that $P = P_1 \cup \cdots \cup P_k$. Moreover, $\phi_0(P) = \phi(P_1) = \cdots = \phi(P_k)$. Denoting by $\psi$ the facet-function associated with $\phi$, this implies for a facet $F \in \mathcal{P}^{n-1}$ that $\psi(F) = 0$ if and only if $F$ is not subset of any facet $F' \in \mathcal{P}_0^{n-1}$.

Let $H$ be a hyperplane as in the statement of the proposition. The above discussion shows that

$$T = \bigcup \{F \in \mathcal{P}_0^{n-1} \mid F \subseteq H\} = \bigcup \{F \in \mathcal{P}^{n-1} \mid F \subseteq H, \, \psi(F) \neq 0\}.$$

Using $S := T$, we obtain a contradiction to Lemma C.8. $\qquad\square$

## D   Detailed Proof of Theorem 4.1

We prove Theorem 4.1 by translating it into the polyhedral world with the help of Proposition 4.5. In the polyhedral world, the following proposition will be useful. Although its statement is well-known in the discrete geometry community, we include a proof for the sake of completeness. The proof makes use of Radon's theorem (compare Edelsbrunner [1987, Theorem 4.1]), stating that any set of at least $n + 2$ points in $\mathbb{R}^n$ can be partitioned into two nonempty subsets such that their convex hulls intersect.

**Proposition D.1.** *Given $p > n + 1$ vectors $(a_i, b_i) \in \mathbb{R}^n \times \mathbb{R}$, $i \in [p]$, there exists a nonempty subset $U \subsetneq [p]$ featuring the following property: there is no $c \in \mathbb{R}^{n+1}$ with $c_{n+1} \geq 0$ and $\gamma \in \mathbb{R}$ such that*

$$
\begin{aligned}
c^T(a_i, b_i) &> \gamma \quad \text{for all } i \in U, \text{ and} \\
c^T(a_i, b_i) &\leq \gamma \quad \text{for all } i \in [p] \setminus U.
\end{aligned}
\tag{8}
$$

*Proof.* Radon's theorem applied to the at least $n + 2$ vectors $a_i$, $i \in [p]$, yields a nonempty subset $U \subsetneq [p]$ and coefficients $\lambda_i \in [0, 1]$ with $\sum_{i \in U} \lambda_i = \sum_{i \in [p] \setminus U} \lambda_i = 1$ such that $\sum_{i \in U} \lambda_i a_i = \sum_{i \in [p] \setminus U} \lambda_i a_i$. Suppose without loss of generality that $\sum_{i \in U} \lambda_i b_i \leq \sum_{i \in [p] \setminus U} \lambda_i b_i$ (otherwise exchange the roles of $U$ and $[p] \setminus U$).

For any $c$ and $\gamma$ that satisfy (8) and $c_{n+1} \geq 0$ it follows that

$$\gamma < c^T \sum_{i \in U} \lambda_i (a_i, b_i) \leq c^T \sum_{i \in [p] \setminus U} \lambda_i (a_i, b_i) \leq \gamma,$$

proving that no such $c$ and $\gamma$ can exist. $\qquad\square$

The idea for proving Theorem 4.1 is the following procedure: we show that any convex CPWL function with $p > n + 1$ pieces can be expressed as integer linear combination of convex CPWL functions with at most $p - 1$ pieces. Then, this can be applied iteratively until we obtain the desired representation. A crucial step for realizing one iteration of this procedure is the following proposition.

**Proposition D.2.** *Let* $f(x) = \max\{a_i^T x + b_i \mid i \in [p]\}$ *be a convex CPWL function defined on* $\mathbb{R}^n$ *with* $p > n + 1$. *Then there exist a subset* $U \subseteq [p]$ *such that*

$$\sum_{\substack{W \subseteq U, \\ |W| \text{ even}}} \max\{a_i^T x + b_i \mid i \in [p] \setminus W\} = \sum_{\substack{W \subseteq U, \\ |W| \text{ odd}}} \max\{a_i^T x + b_i \mid i \in [p] \setminus W\} \qquad (9)$$

*Proof.* Consider the $p > n + 1$ vectors $(a_i, b_i) \in \mathbb{R}^{n+1}$, $i \in [p]$. Choose $U$ according to Proposition D.1. We show that this choice of $U$ guarantees equation (9).

For $W \subseteq U$, let $f_W(x) = \max\{a_i^T x + b_i \mid i \in [p] \setminus W\}$ and consider its extended Newton polyhedron $P_W = \mathcal{N}(f_W) = \text{conv}(\{(a_i, b_i) \mid i \in [p] \setminus W\}) + \text{cone}(\{-e_{n+1}\})$. By Proposition 4.5, equation (9) is equivalent to

$$P_{\text{even}} := \sum_{\substack{W \subseteq U, \\ |W| \text{ even}}} P_W = \sum_{\substack{W \subseteq U, \\ |W| \text{ odd}}} P_W =: P_{\text{odd}},$$

where the sums are Minkowski sums.

We show this equation by showing that for all cost vectors $c \in \mathbb{R}^{n+1}$ it holds that

$$\max\{c^T x \mid x \in P_{\text{even}}\} = \max\{c^T x \mid x \in P_{\text{odd}}\}. \qquad (10)$$

Let $c \in \mathbb{R}^{n+1}$ be an arbitrary cost vector. If $c_{n+1} < 0$, both sides of (10) are infinite. Hence, from now on, assume that $c_{n+1} \geq 0$. Then, both sides of (10) are finite since $-e_{n+1}$ is the only extreme ray of all involved polyhedra.

Due to our choice of $U$ according to Proposition D.1, there exists an index $u \in U$ such that

$$c^T(a_u, b_u) \leq \max_{i \in [p] \setminus U} c^T(a_i, b_i). \qquad (11)$$

We define a bijection $\varphi_u$ between the even and the odd subsets of $U$ as follows:

$$\varphi_u(W) := \begin{cases} W \cup \{u\}, & \text{if } u \notin W, \\ W \setminus \{u\}, & \text{if } u \in W. \end{cases}$$

That is, $\varphi_u$ changes the parity of $W$ by adding or removing $u$. Considering the corresponding polyhedra $P_W$ and $P_{\varphi_u(W)}$, this means that $\varphi_u$ adds or removes the extreme point $(a_u, b_u)$ to or from $P_W$. Due to (11) this does not change the optimal value of maximizing in $c$-direction over the polyhedra, that is, $\max\{c^T x \mid x \in P_W\} = \max\{c^T x \mid x \in P_{\varphi_u(W)}\}$. Hence, we may conclude

$$\max\{c^T x \mid x \in P_{\text{even}}\} = \sum_{\substack{W \subseteq U, \\ |W| \text{ even}}} \max\{c^T x \mid x \in P_W\}$$

$$= \sum_{\substack{W \subseteq U, \\ |W| \text{ even}}} \max\{c^T x \mid x \in P_{\varphi_u(W)}\}$$

$$= \sum_{\substack{W \subseteq U, \\ |W| \text{ odd}}} \max\{c^T x \mid x \in P_W\}$$

$$= \max\{c^T x \mid x \in P_{\text{odd}}\},$$

which proves (10). Thus, the claim follows. $\qquad \square$

With the help of this result, we can now prove Theorem 4.1.

*Proof of Theorem 4.1.* Let $f(x) = \max\{a_i^T x + b_i \mid i \in [p]\}$ be a convex CPWL function defined on $\mathbb{R}^n$. Having a closer look at the statement of Proposition D.2, observe that only one term at the left-hand side of (9) contains all $p$ affine combinations $a_i^T x + b_i$. Putting all other maximum terms on the other side, we may write $f$ as an integer linear combination of maxima of at most $p - 1$ summands. Repeating this procedure until we have eliminated all maximum terms with more than $n + 1$ summands yields the desired representation. $\qquad\square$