# OpenReview forum: "Towards Lower Bounds on the Depth of ReLU Neural Networks"
_NeurIPS.cc/2021/Conference — NeurIPS 2021 Poster_

### Official Review · Reviewer_zAhu · 2021-07-06

**Rating:** 7
**Confidence:** 4

**Summary:**

This paper studies the role of depth in exactly representing real functions by ReLU networks.
Unlike the common ML setting, the focus is on neural nets that are equal in every point to the target function.
Some partial results are given as well as a conjecture that networks of depth k+1 have strictly more expressivity than networks of depth k.

**Ethical Concerns:**

None.

**Limitations And Societal Impact:**

Yes.

**Main Review:**

Pros:
It is an interesting question to study expressivity in the interpolation setting where pointwise equality is required. The main conjecture (that depth k+1 is strictly larger than depth k) is of interest and the partial results serve as a good intro to the kind of polyhedral theory that is likely to be involved in these problems. The paper is generally well written and conveys the questions well (although some improvements are possible: see below). I agree with the authors that  finding connections between well develop fields such as polyhedral combinatorics and integer programing to neural networks could yield interesting insights.

Cons:
The requirement for equality in all points is very strong and less useful in machine learning. One issue is that it is easy to come up with simple functions that cannot be represented in this model by a finite network. In general, it is easy to obtain strong lower bounds by considering functions with a very large number of linear regions and using the ideas of Montufar et al (2014). See also
"Size and Depth Separation in Approximating Benign Functions with Neural Networks" by Vardi et al (2021). Therefore lower bounds in this model may be less informative with respect to the power or limitations of neural networks.
It would be good if the authors could further justify their representational assumption from a machine learning perspective. Perhaps it is related to memorization and interpolation in neural networks.

The authors do not impose any bounds on the width or size of the networks. It might be of interest to consider what happens if one restricts the size of the ReLU networks to be polynomial: arguably networks used in practice have strong limitation on their size. It could be that such a restriction makes the questions raised in this paper more difficult: see Vardi et al.

With respect to the width bound in section 4 it would be nice to have also some kind of a lower bound, or at least an estimate of what the right answer should be.

I've found the first two paragraphs not very informative reiterating common knowledge that does not seem to be very related to the body of the paper. Perhaps the authors could stress more their novel contribution in the first two paragraphs.

Note after rebuttal: I think the paper studies an interesting question and introduces new techniques that might yield interesting insights.
I thereby change my review to accept.

**Time Spent Reviewing:**

8

---

> ### Author Response · Authors · 2021-08-10
> **Response to Reviewer zAhu**
>
> Thanks for your detailed and helpful review.
>
> You are right that exact representation is a very restrictive requirement and there are many functions (namely all non-CPWL functions) that cannot exactly be represented. Still, we think it is a very fundamental question and core problem in the context of understanding the expressive power of neural networks to classify the set of functions that are exactly representable by a certain architecture, independent of direct practical applications.
>
> You mention a very interesting research direction: what happens if the width is bounded to be polynomial? We believe that this fundamentally changes the questions raised in our paper and the methods which can be used to tackle them. We believe that, with this additional requirement, methods from complexity theory need to be employed. Complexity theoretic lower bounds are always challenging to show, but of course, very interesting. Therefore, we see it beyond the scope of this paper, but we will definitely keep it in mind. Thanks for your reference to Vardi et al., which we will include into our discussion in the final version.
>
> You raise the very interesting question whether our bounds are tight, or whether at least some lower bounds on the width for NNs with logarithmic depth can be shown. We do not have a good answer for this question, but we share some thoughts on this in a general comment to all reviewers.
>
> We will take your suggestions concerning the first two paragraphs into account for the final version.

---

### Official Review · Reviewer_gadk · 2021-07-11

**Rating:** 7
**Confidence:** 4

**Summary:**

The authors study the class of functions that can be represented by a fully connected neural network with ReLU activations. First, They conjectured that for any $k\in\mathbb{N}$, $n=2^k$, the function $f_n(x)=\max\{0,x_1,\dots,x_n\}$ cannot be represented by a fully connected network with $k$ hidden layers, and prove this conjecture for $k=2$ under some mild assumption. Second, they proved the class of functions that can be represented by a $(k + 1)$-layer NN is strictly larger than the class of functions that are linear combinations of $2^k$-term max functions. Finally, they provided a bound on the width of the NN required to represent continuous piecewise linear functions.

**Limitations And Societal Impact:**

Yes, the authors have adequately addressed the limitations and potential negative societal impact of their work.

**Main Review:**

Originality: The related works are adequately cited. The novelty of this paper is high. The three main results in this paper, as mentioned in the above summary part,  will certainly help us have a better understating of deep neural networks from a theoretical way. I have checked the technique parts and find that the proofs are solid. I think this is a significant contribution to deep learning immunity. My only concern is that, the proof of Conjecture 1.2 for $k=2$ case, relies on Assumption 2.4, which is not quite elegant. It will be interesting to prove Conjecture 1.2 for $k=2$ case without any further assumptions.

Quality: This paper is technically sound.

Clarity: This paper is clearly written and well organized. I find it easy to follow.

Significance: I think the results in this paper is significant, as explained above.

**Time Spent Reviewing:**

5 hours

---

> ### Author Response · Authors · 2021-08-10
> **Response to Reviewer gadk**
>
> Thanks for your careful reading of the paper including a check of the technical part and your positive review. We would also love to see/find a proof of Assumption 2.4 or an alternative proof of our conjecture that does not use the assumption. One goal of our paper is to attract more researchers to dive into these topics and make it, one day, possible to completely prove (or refute?) our conjecture.

---

### Official Review · Reviewer_oew5 · 2021-07-14

**Rating:** 7
**Confidence:** 3

**Summary:**

It is known from the universal approximation theorem that ReLU networks with one hidden layer can approximate any continuous function on compact sets arbitrarily well. Instead of approximations, this paper investigates  the set of functions that can precisely be described by a network of certain depth. For a given input dimension D, a ReLU network with L=ceil[ log2(D+1) ] (or more) hidden layers (and arbitrary width) can describe the entire set of piecewise continuous functions. Here, new insight is provided for networks with having between 1 and L layers. In particular, under an unproven assumption, the paper shows that L is a strict lower limit for the depth and that adding layers to a network strictly increases the set of describable functions. This is achieved by studying a natural candidate function to require a larger number of hidden layers. The candidate function suggests a conjecture on a nice description of ReLU network functions of finite depth, which is shown to not hold true. Finally, the paper derives a bound on the width and depth of networks that can describe any piecewise linear function with fixed number of linear pieces.

**Limitations And Societal Impact:**

The paper is carefully put together and describes the assumptions of all statements clearly.  As argues in the main part of the review, the paper would benefit from describing the limitations of the used tools to extend the result and to go beyond the partial results reported in the study.

**Main Review:**

The paper is very well-written and the arguments are carefully put together. All proofs are described in detail.  This theoretical paper is mostly interesting from a mathematical viewpoint. With universal approximation at hand, one could argue that a better understanding of the precise description of the set of ReLU functions does not add much from a practical point of view. However, as the authors point out, a better understanding could also be useful for algorithmic advances.

I found the story of the paper intriguing. The conjecture on the lower limit in depth that is necessary to describe all continuous piecewise linear functions on a D-dimensional input space is reduced to proving that the candidate function max{0,x1,x2,,..,xD} cannot be described with less than ceil[ log2(D+1) ] many hidden layers. This statement seems at first like a fairly easy statement to prove or disprove, but appears to be actually quite difficult. Starting from the candidate function, there is natural hope to conjecture that a ReLU network with L hidden layers can only describe linear combinations of maxima of 2^L terms and no more, but this is shown to be false, which adds to the story that the description of ReLU network functions of a certain depth is complicated.

As a result, the paper does not present a complete story by fully characterizing the functions implemented by ReLU networks of fixed depth.

The following describes the progress made in more detail and discusses its limitations:

(i) The conjecture on the tightness of the known lower bound on necessary depth to describe all piecewise linear functions on a D-dimensional input space holds true for networks with two hidden layers, but only under an unproven assumption. That is, the result is limited to three-layer networks and is even incomplete in that case.

(ii) There are maxima of D+1 terms that can be described by a network of depth less than L=ceil[ log2(D+1) ]. This result is interesting, but oneit also disproves a conjecture that was first mentioned in this same paper and which did not previously appear important.

(iii) A bound on the order of depth and width necessary to exactly describe any continuous piecewise linear function with p linear pieces, saying that the required width is polynomial in p with exponent defined by the input dimension. This is a only a small improvement to previously known results.

The main weakness of the paper is that it misses to describe the complications that need to be overcome to extend these limited results. For example, what are the problems to prove the necessary (unproven) assumption for the result in (i)? If it is a natural assumption that should be believed to (probably) hold true for ReLU networks, then why is difficult to show that it does indeed hold? A discussion of the difficulty to prove this critical assumption would make it easier to appreciate the partial result. Similarly, what are the problems of extending the proof to layers with more than two hidden layers?

A strength of the paper is that the proofs of the partial results use nontrivial methods and require a good understanding on the geometrical setting of linear regions. The proof of statement (i) from above is based on mixed integer programming (MIP) with a nontrivial setup of the MIP problem. This part of the paper also introduces so-called H-conforming functions, which form a natural subset of piecewise continuous functions for the study of ReLU networks and this viewpoint could be interesting outside the study in question. The proofs to (ii) and (iii) use theoretical insight on piecewise continuous functions and their associated polyhedra of linear pieces. These proof ideas and techniques provide some insight on their own.

Taken together, the paper presents an intriguing, theoretically interesting question, but only resolves it partially without explaining well why the partial progress is substantial. Since the ideas involved in setting up the statements and proofs are quite interesting by themselves, I tend to support the acceptance of the paper.


*****
 Two small suggestions for potential minor improvement in the presentation:

- Line 66: Better: "By definition, a continuous function is piecewise linear in case ...“
- Line 96: The sketch of the proof only states half the arguments necessary for the proof. It would just cost a single line or maybe two to state the other half: If the specific n+1 term maximum function can be written with k layers, then all n+1-term maximum functions can.


**Time Spent Reviewing:**

10

---

> ### Author Response · Authors · 2021-08-10
> **Response to Reviewer owe5**
>
> Thanks for your detailed and helpful review.
>
> Concerning your point (iii), you say our width bound is “only a small improvement to previously known results”. To the best of our knowledge, there is no width upper bound of the same type (in terms of number of pieces, with logarithmic depth) in previous literature. In case we missed prior work along these lines, we would be grateful to the reviewer if she/he could point us to some papers containing such an upper bound.
>
> Concerning difficulties that need to be overcome to extend our results, we will definitely add a discussion to our final paper.
> The most obvious next goal would be to prove Assumption 2.4. The assumption is intuitive because every breakpoint introduced at other places needs to be cancelled out later. Therefore, it is natural to assume that these breakpoints do not have to be introduced in the first place. However, this intuition does not seem to be enough for a formal proof because it could be the case that additional breakpoints in intermediate steps that are cancelled out later also influence the behavior of the function at other places, where we allow breakpoints in the end.
>
> Extending our result to more layers is infeasible for computational reasons. One could indeed build an analogous MIP for more layers, but this would be way too large to solve with exact arithmetics. A more promising approach would be to find an alternative (non-computational) proof for the 3-layer case, which is then maybe easier to generalize.
>
> Thanks also for the two minor comments, which we will address in the final version.

---

> > ### Comment · Reviewer_oew5 · 2021-08-19
> > **Thank you for your explanations**
> >
> > I thank the authors for their reply and explanations.
> > On (iii), I apologize for the imprecision in my comment on the limitations (and a possibly subjective evaluation). I meant to argue that, next to the bound on the depth, the additional knowledge of the required width in order of the number of linear pieces itself does not provide much significant insight, since it is unclear how sharp the bound is (probably not sharp, I suppose).

---

### Official Review · Reviewer_YGav · 2021-07-16

**Rating:** 5
**Confidence:** 3

**Summary:**

This paper considers the problem of characterizing exact representations for ReLU networks of a given depth, but any width. It was previously shown that the functions represented by ReLU networks of depth logarithmic in dimension is exactly equal to the set of all continuous piecewise linear functions. This work provides some results which suggest that this result is tight. That is, a depth logarithmic in dimension is necessary to exactly represent all piecewise linear functions.

Proposition 1.3 simplifies this conjecture by giving a simple equivalent condition in terms of representing max(0,x_1,\dots,x_n) and proves the conjecture up to dimension 4 under the assumption 2.4 which is unproven. In section 3 it is shown that the set of functions representable by a depth k network is a strict super set of Max(2^k) - i.e, set of all functions which can written as a linear combination of max of 2^k affine functions. This provides further evidence in support of the conjecture.
Section 4 then extends the results of Arora et. al 2018 to provide an upper bound on the width required for a relu network to exactly represent affine function with p pieces. This bound is $p^{O(n^2)}$.

**Limitations And Societal Impact:**

I think a bit more effort needs to be put in to highlight the exact limitations of the work.

**Main Review:**

Disclaimer: I am not an expert on tropical geometry and related topics.

I would first note that this paper is about exact representations. The closure of $\mathsf{ReLU}_n(k)$ with respect to uniform convergence over compacts contains the space of all continuous functions by universal approximation property. This work is about not taking the closure and considering the exact representations. I acknowledge that this is indeed a hard problem.

The scope of the work seems to be very limited. It is not clear why having exact representation is ever useful in practical machine learning. The authors do mention the learning algorithm by Arora et. al 2018, which I believe they should expand on. The results seem a bit weak. The main result in Section 2 which proves the conjecture only upto dimension 4,  conditioned on an unproven assumption. The bounds in Section 4 are super exponential in dimension, which seems very intractable from a computation perspective. It would be great to see some discussions about lower bounds for the width. Practically speaking, if indeed this is tight, what is the use of exact representations at all?

The paper is very well written but I am skeptical about the relevance of this work in a venue like NeurIPS.

**Time Spent Reviewing:**

4 hours

---

> ### Author Response · Authors · 2021-08-10
> **Response to Reviewer YGav**
>
> Thanks for reading and reviewing the paper. While we agree that the paper has no direct practical impact, we disagree with you that this implies a lack of relevance of exact representations.  From our point of view, understanding the class of exactly representable functions is a very natural and fundamental question in the context of advancing the theoretical understanding of neural networks’ great abilities in learning contexts. We also believe that a better theoretical understanding of the representation abilities of NNs, as we provide in our paper, will stimulate further research that makes use of these insights to improve training algorithms and methods to choose the best architecture.
>
> Concerning our results in Section 4, you are right that our bounds are out of practical scope, but it is the best bound one can currently obtain from a theoretical point of view. This makes it interesting for us. You raise the very interesting question regarding whether our bounds are tight, or whether at least some lower bounds on the width for NNs with logarithmic depth can be shown. We do not have a good answer for this question, but we share some thoughts on this in a general comment to all reviewers.

---

> > ### Comment · Reviewer_YGav · 2021-09-10
> > **Thanks for the Response**
> >
> > I thank the authors for the response. I still am not convinced that the results in this work are strong enough and I choose to keep my score the same.

---

### Official Review · Reviewer_Umtt · 2021-07-20

**Rating:** 5
**Confidence:** 4

**Summary:**

EDIT: After reading authors' responses, I decided to keep my score as is.

The paper studies the problem of exact representation of continuous functions via neural networks with the ReLU activation. The literature on the approximation power of neural nets is large, however the paper here delves into the less studied question of whether the class of *exactly* representable functions strictly increases when adding more layers (with no restrictions on size).

The authors want to understand the function classes exactly represented by different architectures and a step towards this direction is to analayze the class of functions captured by a depth-d neural net (without width constraints) and how this class of functions changes as we get to depth-d+1 neural nets.

It is obvious that ReLU nets will output continuous piecewise linear functions (CPWL for short) and a non-trivial fact from previous works is that $\log(n+1)$ hidden layers suffice to represent *any* CPWL in n dimensions, via a ReLU net. Let d is a parameter for the depth and and ReLU(d) is all functions representable exactly via ReLU nets of depth at most d. The paper tries to understand ReLU(d) as d goes from 0 to $\log(n+1)$.

The authors put forth two equivalent conjectures about the relations between the class of functions ReLU(d) for different d. Conjecture 1.1 states that every additional layer will indeed be substantial in terms of the representational capabilities of ReLU nets up to $d\le \log(n+1)$ of course, at which point all CPWL are representable. The authors reformulate this with Conjecture 1.2 that is a simple statement about max functions.

The authors then show a special case of Conj. 1.2 that corresponds to showing that the max function on 5 variables cannot be representted with 2 or 3 hidden layers, with the caveat that they need a certain assumption on the breakpoints of the function represented by any intermediate neuron. Along the same lines,  the authors show that the class ReLU(k) contains more functions than just taking the max on $2^k+1$ variables. To achieve this they use the theory of polyhedral complexes associated with CPWL functions.

Finally, the authors find upper bounds on the sizes of the networks needed for expressing arbitrary CPWL functions with p linear pieces, as given in Theorem 4.4, which basically involves depth O(logn) nets with width growing as $p^{n^2}$.




**Main Review:**

I like the overall theme of understanding the exact representation capabilities of neural networks, escaping the traditional approximation theory viewpoint. As most of the traditional approximation theory results, rely on large widths to approximately construct step functions to approximate a given function, all these techniques cannot work here and new ideas are necessary.

The main ideas in the paper involve how to relate the max functions with output of ReLU nets. Both conjectures provided in the introduction are very plausible (equivalent of course as the authors show) and the authors take some small first steps towards understanding them.

One of the main results is that "there does not exist a 3-layer NN" to compute max on 5 variables. One quick observation here however is that following the notaion in Conj. 1.2 I think there is a shift by +1 that is not correct (is k=2 or k=3 for the statement?). One annoying thing with this result is that it involves a somewhat technical condition on the breakpoints of intermediate neurons which I can't see how to prove. Furthermore, I believe the result is useful but not very surprising. It basically means that computing the maximum between inputs somehow requires a lot of depth in some sense.

The next result is about ReLU(k) being a superset of the class of max functions on $2^k$ variables. To establish this they use a specific construction involving max functions and compositions between max functions that can be written as a ReLU net but not as a max function on several terms. Although the proof is somewhat complicated, still the result seems not as surprising.

The perhaps more interesting part has to do with Section 4 where the authors derive upper bounds for the size of the net to be able to represent CPWL functions.

Overall, the motivation of the work is interesting from a theoretical point of view, however the results presented are quite weak. Section 2 relied on a technical assumption and only proves a very special case of the (plausible) conjecture 1.1, Section 3 is a comparison between very special class of functions. Section 4 has some interesting techniques. I like that the authors have identified the relation to max functions and have built several ways to analyze the exact representation capabilities of NN, however I believe more and stronger results are necessary to make this a solid contribution.

Questions/Future directions to the authors:
- Could most of these results also be stated for recurrent neural networks? As far as I know several important approximation theory results (e.g., Telgarsky's "Benefits of depth in neural networks") can also be viewed for RNNs instead of feedforward nets, and having the analogous theory for RNNs would be interesting, and can strengthen the overall message of your paper.
- To get the separations for the different depth levels and also to separate ReLU(k) from MAX(2^k), the authors identify the max function as a "source of complexity" in some sense. In particular Proposition 3.2 relies on max of max functions. Why not taking this to the extreme? Specifically, is there another way of using repeated compositions of max functions with themselves in order to get a "sufficiently complicated" function? Notice that works that have exploited this repeated compositions trick include the seminal paper by Telgarsky and also follow-ups that extended Telgarsky's results to much broader family of functions using characterizations from dynamical systems (e.g., "Depth-Width Trade-offs for ReLU Networks via Sharkovsky's Theorem"). The "source of complexity" in that case came from oscillations in the input, and not max alternations between the inputs as in your case.
- Another interesting direction would be to get also tradeoffs for the depth/width or other aspects of the architecture needed to exactly represent certain functions. This again will be a counterpoint to many existing approximation theory results that derive depth/width tradeoffs instead of just approximability results.

**Time Spent Reviewing:**

5

---

> ### Author Response · Authors · 2021-08-10
> **Response to Reviewer Umtt**
>
> We are very grateful for your detailed and helpful review.
>
> Concerning the depth-shift: we think the notation within the paper is consistent, and also consistent with previous literature, even if maybe a bit confusing. There is a distinction between the “number of hidden layers” and the number of “layers” (counting input/output). The former denotes the number of inner ReLU layers, which is one less than the number of affine transformations which we also call the depth. For example, Section 2 deals with 3-layer NNs, which contain 2 hidden ReLU layers and are, thus, computing functions in ReLU(2).
>
> Regarding your feeling that our results in Sections 2 and 3 are not “surprising”, let us highlight that our goal was to provide a firm theoretical confirmation of what we intuitively believe to be true. Due to this intuitive appeal, one could indeed say the results are not “surprising”. Still, it is important to know that they are true from a rigorous point of view. Moreover, it turned out that they are surprisingly difficult to show and we see it as one of our main contributions to develop the techniques we use to arrive at our results.
>
> Thanks also for pointing out three additional future directions. All of them sound indeed very interesting. Here some thoughts from our side:
>
> 1. A trivial (and possibly unsatisfying) answer to your question is: by unfolding, every RNN can be seen as a feedforward NN with repetitive structure. Thus, any lower bound on the depth of a feedforward NN is also valid for (unfolded) RNNs. Conversely, our width upper bounds (in Section 4) somehow rely on repetitive maximum computations. Even though this is not an RNN in the classical sense, these NNs are highly structured and possibly share some of the advantages of RNNs.
>
> 2. We fully agree that we see the max function as a source of complexity in ReLU neural networks. This was indeed a strong motivation for our results in Section 3. It would be great to find some formal correspondence between some max function structure and the required number of layers. Also note that the functions from Prop. 3.2 are not only repeated compositions of max functions, but also involve a sum in intermediate steps, which causes additional complexity. Of course, it is a very interesting question whether one can find another way of repeated function compositions (maxima and possibly sums) to achieve strong(er) separation results.
>
> 3. Many of the already existing tradeoff results (by Telgarsky and others) are of the following type: There exists a function that can be *exactly* represented by an NN with a certain architecture, but every representation with another architecture cannot even approximate this function. Thus, from our understanding, this already implies such a tradeoff for the exact setting. Of course, it remains an open question whether even more drastic tradeoffs can be achieved that only hold for the exact setting.
>
> We will add a discussion about these points in the final version.

---

### Decision · Program_Chairs · 2021-09-27

**Decision:**

Accept (Poster)

**Comment:**

Three reviewers recommend accept and two reviewers recommend weak reject. A main criticism is the unclear significance of the partial results and the unclear relevance to practical machine learning. After weighing the discussion and evaluating the submission, I agree with the positive reviewers that the problem formulation, the approach and the tools presented are sufficiently interesting and relevant to the theoretical understanding of neural networks. Hence I am recommending accept. I ask the authors to add the details promised in the discussion in the final version of the paper and to take the reviewers suggestions carefully into consideration when preparing it.